# Unsupervised Document Representation using Partition Word-Vectors Averaging

## Abstract

Learning effective document-level representation is essential in many important NLP tasks such as document classification, summarization, etc. Recent research has shown that simple weighted averaging of word vectors is an effective way to represent sentences, often outperforming complicated seq2seq neural models in many tasks. While it is desirable to use the same method to represent documents as well, unfortunately, the effectiveness is lost when representing long documents involving multiple sentences. One reason for this degradation is due to the fact that a longer document is likely to contain words from many different themes (or topics), and hence creating a single vector while ignoring all the thematic structure is unlikely to yield an effective representation of the document. This problem is less acute in single sentences and other short text fragments where presence of a single theme/topic is most likely. To overcome this problem, in this paper we present P-SIF, a partitioned word averaging model to represent long documents. P-SIF retains the simplicity of simple weighted word averaging, while taking a document's thematic structure into account. In particular, P-SIF learns topic-specific vectors from a document and finally concatenates them all to represent the overall document. Through our experiments over multiple real-world datasets and tasks, we demonstrate P-SIF's effectiveness compared to simple weighted averaging and many other state-of-the-art baselines. We also show that P-SIF is particularly effective in representing long multi-sentence documents. We will release P-SIF's embedding source code and data-sets for reproducing results.

## 1 Introduction

Distributed word embeddings such as word2vec (Mikolov et al., 2013b) have shown their success in representing words as latent continuous features in low-dimensional fixed spaces, which can capture their semantic meaning. These embeddings have shown impressive success for improving the performance of machine learning algorithms such as text classification tasks. The success of these word representations is recently formally explained using the random walk-based latent variable model by (Arora et al., 2016a).

Many approaches such as (Socher et al., 2013; Le & Mikolov, 2014; Ling et al., 2015; Liu et al., 2015a) are proposed which go beyond words to capture the semantic meaning of sentences. These techniques either used simple composition of the word-vectors or sophisticated neural network architectures for sentence representation. Recently, (Arora et al., 2017) proposed smooth inverse frequency of word vector averaging model to embed a sentence. They further improved their embedding by removing the first principal component of the weighted average vectors. However, all these approaches are limited to capturing the meaning of a single sentence and represent the sentence in the same space as words, thus reducing their expressive power. Generally, a longer text contain words from multiple different themes (or topics), and creating a single vector from simple averaging of word-vectors will disregard all the thematic structure. Hence, these techniques are largely unable to capture the semantic meaning of larger pieces of text (phrases, sentences and paragraphs).

To address this problem, in this paper we present a novel document embedding method called partition SIF weighted averaging (P-SIF), which can efficiently embed documents with multiple sentences. P-SIF learns topic-specific vectors from a document and finally concatenates them all to represent the overall document. Thus, P-SIF retains the simplicity of simple weighted word averaging,

while taking a document's thematic structure into account. (Mekala et al., 2017) recently proposed a clustering based word-vector averaging approach for embedding larger text documents. However, they did not provide any justification or experiments to explain the functionality of their embedding approach. This paper provides an analytical explanation about the success of the mentioned representations and also achieves significant improvements over previous embedding techniques on several natural language tasks. Following are the characteristics of P-SIF.

- P-SIF is capable of embedding larger documents with multiple sentences, as it pays attention to the topic/thematic structure of the document.
- P-SIF is simple because it is based on weighted word vectors averaging rather than complicated tensor factorization or neural networks methods.
- P-SIF is an unsupervised method since it is obtained using pre-trained unsupervised word embeddings without using any label information.
- P-SIF outperforms many existing approaches on textual similarity, textual classification and other supervised tasks.

The remaining part of the paper is organized as follows. Section 2 discusses related work in document representations. Section 3 provides motivation for the partition averaging through a qualitative example. Section 4 describes our embedding algorithm. Section 5 introduces and explains the background needed. These sections are followed by experiments in Section 6 and the analysis with the discussion in Section 7.

## 2    RELATED WORK

**Word embeddings.** There are mainly two methodologies proposed for unsupervised word-embedding representations, either by internal representations learned through neural network models of text (Bengio et al., 2003; Collobert & Weston, 2008), and Mikolov et al. (2013b) or by low-rank approximation of co-occurrence statistics by (Levy & Goldberg, 2014; Hashimoto et al., 2016), and Arora et al. (2016a). (Levy & Goldberg, 2014) show that both techniques are equivalent.

**Sentence embeddings.**    Earlier work has computed sentence embedding by coordinate wise vector and matrix based compositional operation over word vectors, e.g., (Levy & Goldberg, 2014) use unweighted averaging of word vectors for representing short phrases, (Singh & Mukerjee, 2015) proposed tfidf-weighted averaging of word vectors to form document vectors, (Socher et al., 2013) proposed a recursive neural network (RNN) defined over a parse tree and trained with supervision. Next, (Le & Mikolov, 2014) proposed *PV-DM* and *PV-DBOW* model which treats each sentence as a shared global latent vector (or pseudo word). Other approaches use seq2seq models such as Recurrent Neural Networks (Mikolov et al., 2010) and Long Short Term Memory (Gers et al., 2002) which can handle long term dependency, hence, capturing the syntax structure. Other neural network models include use of hierarchy and convolution neural networks, such as (Kim, 2014) and (Kalchbrenner et al., 2014). (Wieting et al., 2015) learned paraphrastic sentence embedding by modifying word embeddings based on the supervision from the Paraphrase pairs dataset (PPDB). Recently, a lot of work is harnessing topic modeling (Blei et al., 2003) along with word vectors to learn better word and sentence representations, e.g., TWE (Liu et al., 2015a), NTSG (Liu et al., 2015a), WTM (Fu et al., 2016), w2v-LDA (Nguyen et al., 2015), TV+MeanWV (Li et al., 2016a), LTSG (Law et al., 2017), Gaussian-LDA (Das et al., 2015), Topic2Vec (Niu et al., 2015), Lda2vec (Moody, 2016), and MvTM (Li et al., 2016b).

**Document embeddings.**    (Kiros et al., 2015) proposed skip-thought document embedding vectors which transformed the idea of abstracting the distributional hypothesis from word level to sentence level. (Wieting et al., 2016a) proposed a neural network model which optimizes the word embeddings based on the cosine similarity of the sentence embeddings. (Gupta et al., 2016) proposed a method called Bag of Words Vector (BoWV), which employs a clustering based technique and tf-idf values to form a composite document vector. They represented documents in higher dimensions by using hard clustering over word embeddings. (Mekala et al., 2017) later extended the model to SCDV by using a fuzzy clustering technique and direct idf weighting of word vectors. The learned representations tried to capture a global context of sentence, similar to an n-gram model. Their method outperformed previous state of art on a variety of NLP tasks.

Table 1: Words with their Topic Proportions

| word | 1 | 2 | 3 | 4 | 5 |
|------|-----|-----|-----|-----|-----|
| data | 0.3 | 0.7 | 0.0 | 0.0 | 0.0 |
| interviewing | 0.0 | 0.0 | 0.8 | 0.0 | 0.2 |
| management | 0.0 | 0.0 | 0.8 | 0.0 | 0.2 |
| public | 0.0 | 0.0 | 0.0 | 0.7 | 0.3 |

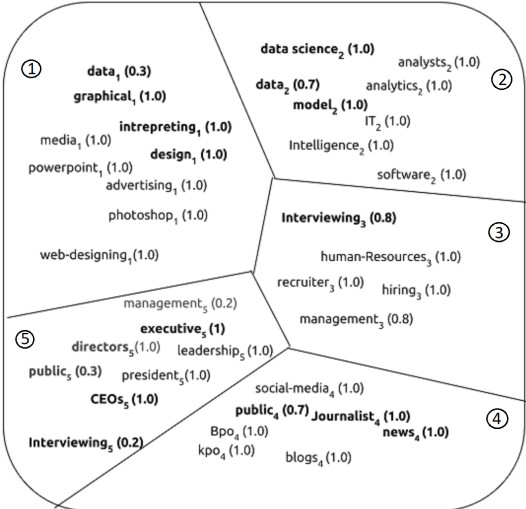

Figure 1: Word vector space for corpus $C$. Words in different partitions are represented by different subscripts and separated by hyperplanes. Bold fonts represent words' presence in the document

## 3 MOTIVATION: AVERAGING VS PARTITION AVERAGING

Let's consider a corpus $(C)$ with $N$ documents with the corresponding most frequent words vocabulary $(V)$. Figure 1, represents the word-vectors space of $V$, where similar meaning words occur closer to each other. We can apply sparse coding to partition the word vectors space to a five (total topics $K = 5$) topic vector space, as shown in figure 1. These five topic vector spaces represent the five topics present in corpus $(C)$. Few words are polysemic and belong to multiple topics with some proportion (see Table 1). In figure 1 we represent the topic number of the word in subscript. Let's consider a document $d_n$: *'Data journalists deliver data science news to general public. They often take part in interpreting the data models. In addition, they create graphical designs and interview the directors and CEO's.'*

One can directly average the word vectors to represent the document $(\vec{v}_{d_n})$, as in (SIF), shown in equation equation 1. Here, $+$ represents element-wise vector addition.

$$\vec{v}_{d_n} = \vec{v}_{\text{data}_2} + \vec{v}_{\text{journalist}_4} + \vec{v}_{\text{news}_4} + \vec{v}_{\text{datascience}_1} + \vec{v}_{\text{public}_4} + \vec{v}_{\text{interpreting}_1} + \vec{v}_{\text{models}_2} +$$
$$\vec{v}_{\text{graphical}_1} + \vec{v}_{\text{design}_1} + \vec{v}_{\text{director}_5} + \vec{v}_{\text{CEO}_5} + \vec{v}_{\text{interviewing}_2} \tag{1}$$

In representation equation 1 it can be seen that we are averaging word vectors which have a very different semantic meaning, e.g., words belonging to partition 1 such as *graphical*, *design*, and *data* are averaged with different semantic meaning words of partition 2 such as *datascience*, *model*, and *data*. In addition, the document is represented in the same $d$ dimensional space as word vectors. Overall, averaging represents the documents as a single point in the vector space and does not consider the $5$ different semantic topics or themes.

However, we do a weighted (topic proportion) average of words within a partition and concatenate the average word vectors across partitions to represent the document $(\vec{v}_{d_n})$, as in (P-SIF), shown in equation equation 2. Here, $+$ represents a element-wise vector addition, $\times$ represents scalar-vector multiplication, and $\oplus$ represents concatenation.

$$\vec{v}_{d_n} = (\vec{v}_{\text{interpreting}_1} + \vec{v}_{\text{graphical}_1} + \vec{v}_{\text{design}_1} + 0.3 \times \vec{v}_{\text{data}_1}) \oplus (0.7 \times \vec{v}_{\text{data}_2} + \vec{v}_{\text{datascience}_2}$$
$$+ \vec{v}_{\text{models}_2}) \oplus (\vec{v}_{\text{journalist}_4} + \vec{v}_{\text{news}_4} + 0.7 \times \vec{v}_{\text{public}_4}) \oplus (\vec{v}_{\text{director}_5} + 0.3 \times \vec{v}_{\text{public}_5} + \vec{v}_{\text{CEO}_5} \quad (2)$$
$$+ 0.2 \times \vec{v}_{\text{interviewing}_5}) \oplus (0.8 \times \vec{v}_{\text{interviewing}_3})$$

In representation equation 2, the final representation takes 5 different semantic topics into account. Words belonging to different semantic topics are separated by concatenation ($\oplus$) as they represent separate meanings, whereas words coming from same topics are simple averages since they represent the same meaning. E.g., average of words belonging to partition 1 such as $graphical$, $design$, and $data$ are concatenated to average of words of partition 2 such as $data - science$, $model$, and $data$. The final document vector $\vec{v}_{d_n}$ is represented in a higher $5 \times d$ dimension vector space, thus having more representational power, where $d$ is the dimension of word vectors. From equation 2, it is clear that the final representation takes the weight to which each word belongs to various topics into account as well, thus handling the multi-sense nature of words, e.g., words such as $data$ belong to partition 1 with probability 0.3 and partition 2 with probability 0.7. Hence, partitioned averaging with topic weighting is essential for representing longer text documents. Thus, P-SIF takes documents' thematic structure into account while doing simple weighted word vectors averaging.

## 4   THE PROPOSED ALGORITHM: P-SIF

In this section, we present the proposed document embedding learning algorithm. We introduce the formal notation needed for the discussion.

- $C$ represents text corpus and $V$ represents vocabulary of words in corpus.
- $\vec{v}_w \in R^d$ represents the word vector of word $w$, where $d$ is dimension of word vector.
- $p(w)$ represents the unigram probability of word $w$ in the corpus.
- $c_0$ and $\vec{v}_{c_0} \in R^d$ represent common context and it's corresponding context vector.

The details to construct document embeddings are given in algorithm 1. The feature formation algorithm can be divided into three major steps :

**Dictionary Learning for Word Vectors:**  Given word vectors $v_w \in R^d$, a sparsity parameter $k$, and an upper bound $m$ , we find a set of unit norm vectors $\vec{A}_1, \vec{A}_2, \ldots, \vec{A}_m$, such that $\vec{v}_w = \sum_{j=1}^{m} \alpha_{(w,j)} \vec{A}_j + \vec{\eta}_w$ , where at most $k$ of the coefficients $\alpha_{(w,1)}, \ldots, \alpha_{(w,m)}$ are nonzero (so-called sparsity constraint), and $\vec{\eta}_w$ is a noise vector.

Sparse coding is usually solved for given $m, k$ by using alternating minimization (Arora et al., 2016b) and (Aharon et al., 2006) to find the $\vec{A}_i'$s that minimizes the following $L_2$-reconstruction error : $\|\vec{v}_w - \sum_{j=1}^{m} \alpha_{(w,j)} \vec{A}_j\|$. Here, $\vec{A}_1, \ldots, \vec{A}_m$ will represents important topic basis in the corpus, which we refer to as the atoms of the topic. Furthermore, restricting $m$ to be much smaller than the number of the words ensures that the same topic needs to be used for multiple words. $\vec{A}_j$ is an interesting or significant topic because the sparse coding ensures that each basis element is softly chosen by many words.

**Word Topics Vector Formation:** For each word $\vec{w}$, we created $K$ different word-cluster vectors of $d$ dimensions $\vec{cv}_{wk}$ by weighting the word embedding with its learned dictionary coefficient $\alpha_{w,k}$ of the $k^{th}$ context.[1] We then concatenated all the $K$ word-cluster vectors $\vec{cv}_{wk}$ into a $K \times d$ dimensional embedding to form a word-topics vector $\vec{tv}_w \in R^{K \times d}$.

$$\vec{cv}_{w,k} \leftarrow \vec{v}_w \times \alpha_{w,k} \quad ; \quad \vec{tv}_w \leftarrow \bigoplus_{k=1}^{K} \vec{cv}_{wk}$$

Here, $\bigoplus$ represents concatenation operation, and $\times$ represents vector-scalar multiplication.

**Smooth Inverse Frequency (SIF) re-weighted:** Finally, for all words appearing in document $D_n$, we weighted the word-topics vectors $\vec{tv}_i$ by smooth inverse frequency $\left(\frac{a}{a+p(w)}\right)$. Next, we removed the common context from the weighted average document vectors by removing the first principal

---

[1]  Empirically, we observed that this weighting generally improves the performance

component from the weighted average vectors. [2]. (Arora et al., 2017) empirically shows that SIF weighting outperforms the tf-idf weighting. They provide a theoretical explanation for their superior performance.

$$\vec{v}_{d_n} \leftarrow \vec{v}_{d_n} - \vec{u}.\vec{u}^T\vec{v}_{d_n}$$

---

**Algorithm 1:** P-SIF Embedding

---

**Data:** Word embeddings $\{\vec{v}_w : w \in V\}$, Documents $\{d_n : d_n \in D\}$, a set of sentences $D$, parameter $a$ and estimated probabilities $\{p(w) : w \in V\}$ of the words, a sparsity parameter $k$, and an upper bound $m$.
**Result:** Document vectors $\{\vec{v}_{d_n} : d_n \in D\}$
  /* Dictionary learning for word-vectors                                */
1 **for** *each word $w$ in $V$* **do**
2    $\vec{v}_w = \sum_{j=1}^{m} \alpha_{w,j}\vec{A}_j + \vec{\eta}_w$;
3 **end**
  /* Word topic-vector formation                                        */
4 **for** *each word $w$ in $V$* **do**
5    **for** *each coefficient, $\alpha_{w,k}$ of word $w$* **do**
6       $\vec{cv}_{w,k} \leftarrow \vec{v}_w \times \alpha_{w,k}$;
7    **end**
8    $\vec{tv}_w \leftarrow \bigoplus_{k=1}^{K} \vec{cv}_{wk}$ ;
      /* $\bigoplus$ is concatenation, $\times$ is scalar vector multiplication      */
9 **end**
  /* SIF reweighed document vector embedding                       */
10 **for** *each document $d_n$ in $D$* **do**
11    $\vec{v}_{d_n} \leftarrow \frac{1}{|d_n|} \sum_{w \in d_n} \frac{a}{a+p(w)} \vec{tv}_w$;
12 **end**
13 Form a matrix $X$ whose columns are $\{\vec{v}_{d_n} : d_n \in D\}$, and let $\vec{u}$ be the first singular vector;
14 **for** *each document $d_n \in D$* **do**
15    $\vec{v}_{d_n} \leftarrow \vec{v}_{d_n} - \vec{u}\vec{u}^T\vec{v}_{d_n}$ ;
16 **end**

---

## 5 P-SIF DISCUSSIONS

For single sentence documents all words of a document belong to a single topic. However, for multiple sentence-documents, words of a document generally originate from multiple topics. To capture this phenomenon, topic modeling algorithms such as LDA (Blei et al., 2003) are used to represent the documents. These representations essentially represent the global contexts of the documents as a distribution over topics. However, these representations do not take the local context initiating from the distributional semantics into account such as word vectors. To consider both local and global contexts, we represented each word as a word-topic-vector $\vec{tv}_w$. In this section, we describe our simple and efficient unsupervised method for document representation in details.

**Sparse Dictionary Learning (Algorithm 1: Lines 1 - 3):** (Arora et al., 2016b) shows that atoms of sparse coding over word-vectors represent all prominent topics in the corpus. Furthermore, they showed that multiple word senses of a word reside as a linear superposition within the word embedding and can be recovered by simple sparse coding. Therefore, one can use the sparse coding of word vectors to detect multiple senses of words and total senses of all the words (number of basis). To find these topics, we used sparse coding algorithms such as k-svd (Aharon et al., 2006) over word vectors $\vec{v}_w$. For a given word ($w$), the $k$ non-zero coefficient essentially represents the distribution of words over topics. The $k$ non-zero $\alpha_w$ for a given word $w$, basically represents the multi-sense nature of the words.

**Word Topics Vector (Algorithm 1: Lines 3 - 9):** Since our multi-sentence documents have words coming from multiple topics, we did not directly average word-vectors with a simple averaging technique. We concatenated the word embeddings over the topic distribution of the words. This helps to represent semantically similar words in the same topic, while words which are semantically

---

[2] We did not remove the common component from final vectors, when we used Doc2VecC (Chen, 2017) initialized word vectors with P-SIF, because frequent words' word-vectors become close to $\vec{0}$

different are represented in different topics. Concatenation of word embeddings over topics also helps in the expression of the multi sense nature of the words. We weigh word-vectors by coefficients of the learned dictionary to capture the cross correlation $(\alpha_i \alpha_j)$ between contexts.

**SIF Weight Averaging and Common Component Removal (Algorithm 1: Lines 9 - 16):** Instead of assuming a single topic for the whole document, we showed that the total number of topics over a given corpus is $K$ (as shown by (Arora et al., 2016b)) and for a given word, $k$ out of $K$ of them would be active. Compared to SIF which directly averages SIF weighted words-vectors, we first partitioned according to the topics through dictionary learning over word-topics vectors. We finally averaged SIF weighted word-topics vectors. Lastly, we removed the first principal components from the document vectors to remove the common component.

**Sparse Dictionary Learning vs. Fuzzy Clustering.** Sparse coding can also be treated as a linear algebraic analogue of overlapping clustering, where the $\vec{A_i}$'s act as cluster centers and each $\vec{v}_w$ is assigned to each cluster in a soft way (using the coefficients $\alpha_{(w,j)}$, of which only $k$ are nonzero) to a linear combination of at most $k$ clusters. In practice, sparse coding optimization produces coefficients $\alpha_{(w,j)}$ which are almost all positive, even though *unconstrained*. One can use fuzzy clustering where each word belongs to every cluster with some probability $P(c_k|w_i)$, which can be thought as a substitute for $\alpha_{(w,k)}$, similar to the approach in SCDV (Mekala et al., 2017). Practically, when the number of contexts is large, dictionary learning performs better than fuzzy clustering due to 1) better optimization and non-redundant clusters and 2) automatic handling of the tail of $P(c_k|w_i)$ distribution through sparsity constraints (Olshausen & Field, 1997; Gao et al., 2010; Yang et al., 2009). We observed that for a single sentence document with a small number of topics it is better to use fuzzy clustering because it has a better optimization and learns efficient clusters. Additionally, in fuzzy clustering, the $P(c_k|w_i)$ is a probability between 0 and 1, which sums to 1 over all contexts. However, to overcome the noise from the non-zero tail probability one can apply hard thresholding on the final vectors, which are directly learned by few non-zero coefficients $(\alpha_{w,j})$ of the dictionary.

Our approach is averaging the words which belong to similar topics and concatenating these averages across different topics. Therefore, our approach is a strict generalization of the sentence embedding approach by (Arora et al., 2017) which is a special case where total topics or themes is $K = 1$.

# 6 EXPERIMENTAL RESULTS

We performed several experiments on several text similarity and classification tasks. We will address the following research questions through our experiments

Q1. Why partition and sparsity is required during word vectors averaging for representing documents?

Q2. Does P-SIF represent large text documents better compared to other techniques used for representing a single sentence?

Q3. What is the effect of SIF weighting and common component removal on document representation?

## 6.1 TEXTUAL SIMILARITY TASK

**Datasets.** We performed our experiments on the SemEval dataset (2012-2017). These experiments involved 27 semantic textual similarity (STS) tasks (2012 - 2016) (Agirre et al., 2012; 2013; 2014; 2015; 2016), the SemEval 2015 Twitter task (Xu et al., 2015), and the SemEval 2014 Semantic relatedness task (Marelli et al., 2014). The objectives of these tasks are to predict the similarity between two sentences. We used the Pearsons coefficient (Fleiss & Cohen, 1973) between the predicted scores and the ground-truth scores for the evaluation. Please refer to the supplementary section F.1 for the experimental setting details.

1. **Unsupervised**: We used ST, avg-Glove, tfidf-Glove and Glove + WR as a baseline. ST denotes the skip-thought vectors by (Kiros et al., 2015), avg-Glove denotes the unweighted average of the Glove Vectors by (Pennington et al., 2014b) [3], and tfidf-Glove denotes the tf-

---

[3] We used the 300-dimensional word vectors that are publicly available at http://nlp.stanford.edu/projects/glove/

Table 2: Experimental results (Pearsons r × 100) on textual similarity tasks. The highest score in each row is in bold. See the experiment settings section 6 under the textual similarity task for the description of the methods. Many results are collected from (Wieting et al., 2016a) and (Wieting & Gimpel, 2017) (GRAN) except tfidf-GloVe and our new representation

| Supervised or not | Supervised | | | | | | | | UnSupervised | | | Semi Supervised | | | P-SIF |
|---|---|---|---|---|---|---|---|---|---|---|---|---|---|---|---|
| Tasks | PP | PP-proj | DAN | RNN | iRNN | LSTM (no) | LSTM (o.g.) | GRAN | ST | avg Glove | tfidf Glove | avg PSL | Glove +WR | PSL +WR | P-SIF +PSL |
| STS12 | 58.7 | 60.0 | 56.0 | 48.1 | 58.4 | 51.0 | 46.4 | 62.5 | 30.8 | 52.5 | 58.7 | 52.8 | 56.2 | 59.5 | **65.70** |
| STS13 | 55.8 | 56.8 | 54.2 | 44.7 | 56.7 | 45.2 | 41.5 | 63.4 | 24.8 | 42.3 | 52.1 | 46.4 | 56.6 | 61.8 | **63.98** |
| STS14 | 70.9 | 71.3 | 69.5 | 57.7 | 70.9 | 59.8 | 51.5 | **75.9** | 31.4 | 54.2 | 63.8 | 59.5 | 68.5 | 73.5 | 74.80 |
| STS15 | 75.8 | 74.8 | 72.7 | 57.2 | 75.6 | 63.9 | 56.0 | **77.7** | 31.0 | 52.7 | 60.6 | 60.0 | 71.7 | 76.3 | 77.29 |
| SICK14 | 71.6 | 71.6 | 70.7 | 61.2 | 71.2 | 63.9 | 59.0 | 72.9 | 49.8 | 65.9 | 69.4 | 66.4 | 72.2 | 72.9 | **73.41** |
| Twitter15 | 52.9 | 52.8 | 53.7 | 45.1 | 52.9 | 47.6 | 36.1 | 50.2 | 24.7 | 30.3 | 33.8 | 36.3 | 48.0 | 49.0 | **54.86** |

Table 3: Experimental results (Pearsons r × 100) on textual similarity tasks on STS 16. The highest score is in bold.

| Tasks | Skip thoughts | LSTM | Tree LSTM | Sent2Vec | Doc2Vec | Glove Avg | Glove tf-idf | PSL Avg | PSL tf-idf | Glove +WR | PSL +WR | P-SIF +PSL |
|---|---|---|---|---|---|---|---|---|---|---|---|---|
| STS16 | 51.4 | 64.9 | 64.0 | **73.7** | 69.4 | 47.2 | 51.1 | 63.3 | 66.9 | 72.4 | 72.5 | **73.7** |

idf weighted average of Glove vectors. We also compared our method with the SIF weighting ($W$) common component removal ($R$) Glove vectors (Glove + $WR$) by (Arora et al., 2017). For STS 16 we also compared our embedding with Skip-Thoughts (Kiros et al., 2015) and Sent2Vec (Pagliardini et al., 2018) embeddings.

2. **Semi-Supervised**: We used avg-PSL, PSL + WR, and the avg-PSL used the unweighted average of the PARAGRAM-SL999 (PSL) word vectors by (Wieting et al., 2015) as a baseline, obtained by training on PPDB dataset(Ganitkevitch et al., 2013). The word vectors are trained using unlabeled data. Furthermore, Sentence embeddings are obtained from unweighted word vectors averaging. We also compared our method with the SIF weighting (W) common component removal (R) PSL word vectors (PSL + WR) by (Arora et al., 2017).

3. **Supervised**: We compared our method with PP, PP-proj., DAN, RNN, iRNN, LSTM (o.g), LSTM(no) and GRAN. All these methods are initialized with PSL word vectors and then trained on PPDB dataset (Ganitkevitch et al., 2013). PP(Wieting et al., 2016a) is the average of word vectors while PP-proj is the average of word vectors followed by a linear projection. The word vectors are updated during the training. DAN denotes the deep averaging network of (Iyyer et al., 2015). RNN is a Recurrent neural network, iRNN is identity activated Recurrent Neural Network based on identity initialize weight matrices. The LSTM is the version from (Gers et al., 2002), either with output gates (denoted as LSTM (o.g.)) or without (denoted as LSTM (no)). GRAN denotes the state of art supervised averaging based Gated Recurrent Averaging Network from (Wieting & Gimpel, 2017). For STS 16 we also compared our embedding with Tree-LSTM (Tai et al., 2015) embedding.

More details on the values of the hyper-parameters used for the experiments are described in details in the supplementary section F.1.

**Results and Analysis** The average results for each year are reported in Table 2. We denoted our embeddings by P-SIF + PSL (+ PSL denotes using the PSL word vectors). We reported the average results for the STS tasks. The detailed results on each sub-dataset are in the supplementary section A. We observed that P-SIF + PSL outperforms PSL + WR on all datasets, thus supporting the usefulness of our partitioned averaging. Our method also outperformed Neural Network models such as LSTM and RNN. Our proposed techniques performed the best over 16 datasets out of 22 compared to methods from (Wieting et al., 2016a) and (Arora et al., 2017) (refer to supplementary section A for details). Our method outperformed supervised averaging based Gated Recurrent Averaging Network (GRAN) on 11 datasets. Furthermore, our results outperformed recently pro-

posed unsupervised methods such as Skip-Thoughts(Kiros et al., 2015), Sent2Vec(Pagliardini et al., 2018), and supervised method such as Tree-LSTM(Tai et al., 2015). We observed that partitioning through fuzzy cluster algorithm such as GMM generates a better performance compared to partitioning through sparse dictionary algorithms such as k-svd for most Semantic Textual Similarity (STS) tasks. The main reason for this peculiar observation was related to the fact that the STS datasets contains documents which are single sentences of a maximum length of 40 words. As discussed in section 5 (sparse dictionary learning vs. fuzzy clustering), for single sentence documents with a small number of topics, fuzzy clustering optimizes better than sparse dictionary learning. Therefore, we used GMM for the STS task which results in partitioning words into suitable clusters. Both ksvd and GMM outperform SIF, however, the improvement was more observable with the GMM-based partitioning. It should be noted that this observation does not hold for 20NewsGroup and Reuters datasets since the documents in these datasets have multiple sentences with a total number of words $>> 40$. We also report some qualitative results in the supplementary section D with a supporting example in section E. In addition, we report our results on the SICK supervised classification task in supplementary section B.

## 6.2 TEXTUAL CLASSIFICATION TASK

The document embeddings obtained by our method can be used as direct features for downstream many supervised tasks.

**Datasets** We ran multi-class experiments on 20NewsGroup dataset [4] and multi-label classification experiments on Reuters-21578 dataset [5]. We used *script* [6] for preprocessing the dataset. Please refer to supplementary section F.1 for the hyperparameter's details.

**Baselines** We considered the following baselines: The Bag-of-Words (BoW) model ((Harris, 1954)), the Bag of Word Vector (BoWV) (Gupta et al., 2016) model, Sparse Composite Document Vector (SCDV) (Mekala et al., 2017) [7] paragraph vector models (Le & Mikolov, 2014), Topical word embeddings (TWE-1) (Liu et al., 2015b), Neural Tensor Skip-Gram Model (NTSG-1 to NTSG-3) (Liu et al., 2015a), tf-idf weighted average word-vector model(Singh & Mukerjee, 2015) and weighted Bag of Concepts (weight-BoC) (Kim et al., 2017) where we built document-topic vectors by counting the member words in each topic, and Doc2VecC (Chen, 2017) where averaging and training of word vectors are done jointly. Moreover, we used SIF (Arora et al., 2017) smooth inverse frequency weight with common component removal from weighted average vectors as a baseline. We also compared our results with other topic modeling based document embedding methods such as WTM (Fu et al., 2016), w2v-LDA (Nguyen et al., 2015), LDA (Chen & Liu, 2014), TV+MeanWV (Li et al., 2016a)), LTSG (Law et al., 2017), Gaussian-LDA (Das et al., 2015), Topic2Vec (Niu et al., 2015), Lda2Vec (Moody, 2016) and MvTM (Li et al., 2016b). Please refer to supplementary section F.2 for the hyperparameter's details.

Table 4: Performance on multi-label classification on Reuters. P-SIF represents our new embeddings. P-SIF(Doc2VecC) represents embeddings obtained using Doc2VecC-initialized word-vectors. Values in bold show the best performance.

| Model | Prec@1 nDCG@1 | Prec @5 | nDCG @5 | Coverage Error | LRAPS | F1-Score |
|---|---|---|---|---|---|---|
| P-SIF (Doc2VecC) | **94.92** | **37.98** | **50.40** | **6.03** | **93.95** | **82.87** |
| P-SIF | **94.77** | **37.33** | **49.97** | **6.24** | **93.72** | **82.41** |
| SCDV(Mekala et al., 2017) | 94.20 | 36.98 | 49.55 | 6.48 | 93.30 | 81.75 |
| Doc2VecC(Chen, 2017) | 93.45 | 36.86 | 49.28 | 6.83 | 92.66 | 81.29 |
| BoWV(Gupta et al., 2016) | 92.90 | 36.14 | 48.55 | 8.16 | 91.46 | 79.16 |
| TWE-1(Liu et al., 2015b) | 90.91 | 35.49 | 47.54 | 8.16 | 91.46 | 79.16 |
| SIF(Arora et al., 2017) | 90.40 | 35.09 | 47.32 | 8.98 | 88.10 | 76.78 |
| PV-DM(Le & Mikolov, 2014) | 87.54 | 33.24 | 44.21 | 13.15 | 86.21 | 70.24 |
| PV-DBoW(Le & Mikolov, 2014) | 88.78 | 34.51 | 46.42 | 11.28 | 87.43 | 73.68 |
| AvgVec(Singh & Mukerjee, 2015) | 89.09 | 34.73 | 46.48 | 9.67 | 87.28 | 71.91 |
| tfidf AvgVec(Singh & Mukerjee, 2015) | 89.33 | 35.04 | 46.83 | 9.42 | 87.90 | 71.97 |

---

[4] http://qwone.com/$\sim$jason/20Newsgroups/     [5] https://goo.gl/NrOfu
[6] https://gist.github.com/herrfz/7967781   [7] https://github.com/dheeraj7596/SCDV

**Multi-class classification** We evaluated the classifier's performance using standard metrics such as accuracy, macro-averaging precision, recall and F-measure. Table 5 shows a comparison on multiple state-of-art document representations on the 20NewsGroup dataset.

Table 5: Performance on multi-class classification on 20NewsGroups. P-SIF represents our new embeddings. P-SIF (Doc2VecC) represents embeddings obtained from Doc2VecC-initialized word-vectors. Values in bold show the best performance.

| Model | Acc | Prec | Rec | F-mes |
|---|---|---|---|---|
| P-SIF (Doc2VecC) | **86.0** | **86.1** | **86.1** | **86.0** |
| P-SIF | **85.4** | **85.5** | **85.4** | **85.2** |
| SCDV(Mekala et al., 2017) | 84.6 | 84.6 | 84.5 | 84.6 |
| Doc2VecC(Chen, 2017) | 84.0 | 84.1 | 84.1 | 84.0 |
| BoE(Jin et al., 2016) | 83.1 | 83.1 | 83.1 | 83.1 |
| NTSG-1(Liu et al., 2015a) | 82.6 | 82.5 | 81.9 | 81.2 |
| NTSG-2(Liu et al., 2015a) | 82.5 | 83.7 | 82.8 | 82.4 |
| SIF(Arora et al., 2017) | 82.3 | 82.6 | 82.9 | 82.2 |
| BoWV(Gupta et al., 2016) | 81.6 | 81.1 | 81.1 | 80.9 |
| NTSG-3(Liu et al., 2015a) | 81.9 | 83.0 | 81.7 | 81.1 |
| LTSG(Law et al., 2017) | 82.8 | 82.4 | 81.8 | 81.8 |
| WTM(Fu et al., 2016) | 80.9 | 80.3 | 80.3 | 80.0 |
| w2v-LDA(Nguyen et al., 2015) | 77.7 | 77.4 | 77.2 | 76.9 |
| TV+MeanWV(Li et al., 2016a) | 72.2 | 71.8 | 71.5 | 71.6 |
| MvTM(Li et al., 2016b) | 72.2 | 71.8 | 71.5 | 71.6 |
| TWE-1(Liu et al., 2015b) | 81.5 | 81.2 | 80.6 | 80.6 |
| Lda2Vec(Moody, 2016) | 81.3 | 81.4 | 80.4 | 80.5 |
| LDA (Das et al., 2015) | 72.2 | 70.8 | 70.7 | 70.0 |
| weight-AvgVec(Singh & Mukerjee, 2015) | 81.9 | 81.7 | 81.9 | 81.7 |
| BoW(Harris, 1954) | 79.7 | 79.5 | 79.0 | 79.0 |
| weight-BOC(Kim et al., 2017) | 71.8 | 71.3 | 71.8 | 71.4 |
| PV-DBoW(Le & Mikolov, 2014) | 75.4 | 74.9 | 74.3 | 74.3 |
| PV-DM(Le & Mikolov, 2014) | 72.4 | 72.1 | 71.5 | 71.5 |

**Multi-label classification:** We evaluated multi-label classification performance using Precision@K, nDCG@k(Bhatia et al., 2015), Coverage error, Label ranking average precision score (LRAPS) [8] and F1-score. Table 4 shows the evaluation results for multi-label text classification on the Reuters-21578 dataset.

**Results and Analysis** We observed that P-SIF outperforms all other methods by a significant margin on both 20NewsGroup and Reuters datasets. We observed that the dictionary learns more diverse and non-redundant topics compared to fuzzy clustering (SCDV) since we only require 40 partitions rather than 60 partitions in SCDV to obtain the best performance. Simple tf-idf weighted averaging based document representation does not show significant improvement in performance by increasing word vector dimensions. We achieved a $< 0.4$ % improvement in accuracy when the word vectors dimensions increase from 200 to 500 on 20NewsGroup. We observed that increasing word vectors dimensions beyond 500 does not improve the performance of SIF and P-SIF. We further improved the performance on both datasets using Doc2vecC initialized word-vectors with the help of the P-SIF algorithm. We represented this approach by P-SIF (Doc2VecC) in Table 5 and Table 4. On 20NewsGroup we required only 20 partitioned instead of 40 with Doc2VecC initialized word vectors. This shows that better word vector representations help in learning more diverse and non-redundant partitions. We also reported our results on each of the 20 classes of 20NewsGroup separately in the supplementary section C.

## 7   ANALYSIS AND DISCUSSION

**Effect of Document-Length:** We conducted a small experiment to show that our model performs better compared to SIF when we have large size documents. We have divided 26 STS (sentence textual similarity) datasets by average document length, i.e., number of words in documents in bins of (10-20, 20-30, 30-40, 40-50) words. Next, we averaged the relative performance improvement in accuracy with respect to SIF $\left(\frac{\text{P-SIF}-SIF}{SIF}\%\right)$ and over all datasets in each bin. In Figure 2, we observe that for complex multi-sentence documents with more words P-SIF relatively performs better. We have also observed that short texts require fewer number of partitions to achieve their best performance which was quite intuitive since short text documents will map into fewer topics.

---

[8] https://goo.gl/4GrR3M

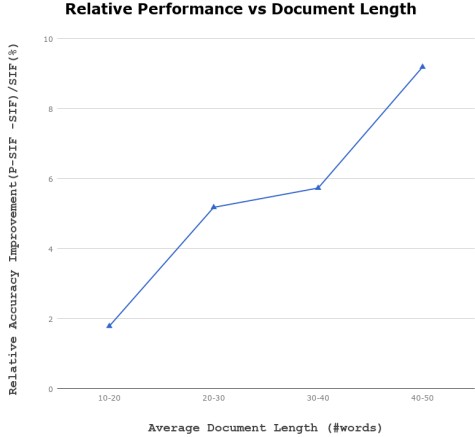

Figure 2: Relative performance improvement pf P-SIF over SIF in accuracy $\left( \frac{\text{P-SIF} - SIF}{SIF} \% \right)$ w.r.t average numbers of words in documents for 27 sts (sentence textual similarity) datasets.

**Effect of Partition:** We have concatenated the word embeddings over topic distribution of words which assigned semantically similar words to identical partitions and semantically different words to different partitions. Partitioning and concatenation of word embeddings over topics also helps in the representation of multi-sense words, which would have been left-out by simple averaging of the word embeddings in document representation otherwise. We represented our words by partitioned average of word-vector embeddings distributed over partitions, which grew the representation by the multiplication of the number of partitions, compared to word vector embeddings. Such high dimensional data structure regularizers, e.g., sparse encodings, help in overcoming the curse of dimensionality. We used k-svd (Aharon et al., 2006) a generalized k-means iterative method that alternates between the sparse coding of the examples based on the current dictionary and updating the dictionary atoms to fit the data. SCDV (Mekala et al., 2017) does manual sparse-encoding to ensure sparsity. (Mekala et al., 2017) clustering methodology uses GMM for partitioning the vocabulary space and capturing the topics. Instead of GMM, we used a dictionary learning based approach which imposes a sparsity constraint implicitly during optimization. Since, k-svd is internally and alternatively optimized between sparse encodings and topic representations, we are able to solve the curse of higher dimensionality by automatic sparse representations of the partitions. Emperically, on both datasets, we observed that the dictionary learns more diverse and non-redundant topics compared to fuzzy clustering. We required only $40$ partitions rather than $60$ in SCDV to obtain the best performance. Additionally, we achieved the best performance with (P-SIF(Doc2VecC)) with just $(20 * 300)$ dimensions of word embeddings (mostly sparse) as compared to $(60 * 300)$ dimensions of word-embeddings (mostly non-sparse) in SCDV. Thus, we obtained a performance gain of $1.5\%$ with 0.33 of the size of the SCDV embeddings.

**Effect of SIF and Common Component Removal:** We have also shown improvement in shorter text tasks via sentence textual similarity task. We have used SIF based weighted word averaging methods instead of tf-idf averaging (Mekala et al., 2017) or simple averaging of word-embeddings. SIF takes advantage of the generative process of document-formation and represents documents as simple smooth inverse frequency averaging of word embeddings. We also observe that removing the first principal component helps in removing the common component occurring due to frequent common words from document vectors. This observation reduces noise and redundancy from the composed document vectors which make vectors more discriminating.

## 8 CONCLUSIONS AND FUTURE WORK

We proposed an unsupervised document feature formation technique based on a partitioned word vector averaging method. Our embedding retains the simplicity of simple weighted word averaging, while taking documents' thematic structure into account. Our simple and efficient approach achieved significantly better performance on several textual similarity tasks, textual classification tasks and

out of domain tasks. In the future, three extensions of our work would be 1. Learning a supervised weighting scheme of words according to the task to improve words weighting, 2. Learning and using context sensitive word embeddings instead of normal skip-gram embeddings to resolve cluster disambuity, and 3. Learning a lower dimensional space manifold to represent word-topic vectors for continuous representations useful in deep leaning applications.

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

# Supplementary: Unsupervised Document Representation using Partition Word-Vectors Averaging

## A    UNSUPERVISED TASK: TEXTUAL SIMILARITY

In this supplementary section, we present the details results for STS tasks for each year. Each year, there are 4 to 6 STS tasks, as shown in Table 6. Note that tasks with the same name in different years are different tasks in reality. We provide detailed results for each tasks in STS 12 - 15 in Table 7. Our method outperforms all other methods from (Arora et al., 2017) and (Wieting et al., 2016a) on all 16 out of 22 tasks. Our method performs significantly better in comparison to all unsupervised embedding methods. In addition, P-SIF is very close to the best performance by supervised methods on the rest of the datasets. The performance difference between best method and our method compared with best method is much small. Our method was also able to outperform state of art supervised averaging based Gated Recurrent Averaging Network (GRAN) (Wieting & Gimpel, 2017) on 11 datasets shown in Table 7. Our results also outperform state of art methods on many recent supervised embedding methods on STS 16 task, as shown in Table 8.

Table 6: The STS tasks by year. Tasks with the same name in different years are different tasks

| STS12 | STS13 | STS14 | STS15 | STS16 |
|---------|----------|-----------|------------------|-------------------|
| MSRpar | headline | deft forum | anwsers-forums | headlines |
| MSRvid | OnWN | deft news | answers-students | plagiarism |
| SMT-eur | FNWN | headline | belief | posteditng |
| OnWN | SMT | images | headline | answer-answer |
| SMT-news | | OnWN | images | question-question |
| | | tweet news | | |

## B    OTHER SUPERVISED TASKS

We also consider three out of domain supervised tasks: the SICK similarity task, the SICK entailment task, and the Stanford Sentiment Treebank (SST) binary classification task by (Socher et al., 2013). We use the setup similar to (Wieting et al., 2016a) and (Arora et al., 2017) for a fair comparison, including the linear projection maps which take the embedding into 2400 dimension (same as skip-thought vectors), and is learned during the training. We compare our method to PP, DAN, RNN, LSTM, and skip-thoughts and other baselines, details results in Table 9.

**Results and Analysis.** Our method (P-SIF) gets better performance compared to PSL + WR on all the three tasks similarity, entailment and sentiment. We obtained best results for two of the supervised tasks, despite many of these methods (DAN, RNN, LSTM) are trained with supervision. Furthermore, the skip thought vectors use has a higher dimension of 2400 dimension instead of 300 dimensions (which we projected to 2400 for a fair comparison). Our method wasn't able to outperform sentiment task compared to supervised tasks because a) due to antonym problem word vectors capture sentimental meaning of words and b) also in our weighted average scheme, we didn't give more weights to sentiment words such as *not*,*good*, *bad*, there may be some important sentiment words which are down-weighted by SIF weighting scheme. However, we outperform PSL + WR by a significant margin and have lesser performance gap with the best supervised approach.

Table 7: Experimental results (Pearsons r × 100) on textual similarity tasks. The highest score in each row is in boldface. The methods can be supervised (denoted as Su.), semi-supervised (Se.), or unsupervised (Un.). See main text for description of the methods. Many results are collected from (Wieting et al., 2016a) and (Wieting & Gimpel, 2017) (GRAN) except tfidf-GloVe and our representation.

| Supervised or not | Supervised | | | | | | | | UnSupervised | | | Semi Supervised | | | P-SIF |
|---|---|---|---|---|---|---|---|---|---|---|---|---|---|---|---|
| Tasks | PP | PP -proj | DAN | RNN | iRNN | LSTM (no) | LSTM (o.g.) | GRAN | ST | avg Glove | tfidf Glove | avg PSL | Glove +WR | PSL +WR | P-SIF +PSL |
| MSRpar | 42.6 | 43.7 | 40.3 | 18.6 | 43.4 | 16.1 | 9.3 | 47.7 | 16.8 | 47.7 | 50.3 | 41.6 | 35.6 | 43.3 | **52.4** |
| MSRvid | 74.5 | 74.0 | 70.0 | 66.5 | 73.4 | 71.3 | 71.3 | 85.2 | 41.7 | 63.9 | 77.9 | 60.0 | 83.6 | 84.1 | **85.6** |
| SMT-eur | 47.3 | 49.4 | 43.8 | 40.9 | 47.1 | 41.8 | 44.3 | 49.3 | 35.2 | 46.0 | 54.7 | 42.4 | 49.9 | 44.8 | **58.7** |
| OnWN | 70.6 | 70.1 | 65.9 | 63.1 | 70.1 | 65.2 | 56.4 | 71.5 | 29.7 | 55.1 | 64.7 | 63.0 | 66.2 | 71.8 | **72.2** |
| SMT-news | 58.4 | **62.8** | 60.0 | 51.3 | 58.1 | 60.8 | 51.0 | 58.7 | 30.8 | 49.6 | 45.7 | 57.0 | 45.6 | 53.6 | 59.5 |
| STS12 | 58.7 | 60.0 | 56.0 | 48.1 | 58.4 | 51.0 | 46.4 | 62.5 | 30.8 | 52.5 | 58.7 | 52.8 | 56.2 | 59.5 | **65.7** |
| headline | 72.4 | 72.6 | 71.2 | 59.5 | 72.8 | 57.4 | 48.5 | **76.1** | 34.6 | 63.8 | 69.2 | 68.8 | 69.2 | 74.1 | 75.7 |
| OnWN | 67.7 | 68.0 | 64.1 | 54.6 | 69.4 | 68.5 | 50.4 | 81.4 | 10.0 | 49.0 | 72.9 | 48.0 | 82.8 | 82.0 | **84.4** |
| FNWN | 43.9 | 46.8 | 43.1 | 30.9 | 45.3 | 24.7 | 38.4 | **55.6** | 30.4 | 34.2 | 36.6 | 37.9 | 39.4 | 52.4 | 54.8 |
| SMT | 39.2 | 39.8 | 38.3 | 33.8 | 39.4 | 30.1 | 28.8 | 40.3 | 24.3 | 22.3 | 29.6 | 31.0 | 37.9 | 38.5 | **41.0** |
| STS13 | 55.8 | 56.8 | 54.2 | 44.7 | 56.7 | 45.2 | 41.5 | 63.4 | 24.8 | 42.3 | 52.1 | 46.4 | 56.6 | 61.8 | **64.0** |
| deft forum | 48.7 | 51.1 | 49.0 | 41.5 | 49.0 | 44.2 | 46.1 | **55.7** | 12.9 | 27.1 | 37.5 | 37.2 | 41.2 | 51.4 | 53.2 |
| deft news | 73.1 | 72.2 | 71.7 | 53.7 | 72.4 | 52.8 | 39.1 | **77.1** | 23.5 | 68.0 | 68.7 | 67.0 | 69.4 | 72.6 | 75.2 |
| headline | 69.7 | 70.8 | 69.2 | 57.5 | 70.2 | 57.5 | 50.9 | **72.8** | 37.8 | 59.5 | 63.7 | 65.3 | 64.7 | 70.1 | 70.2 |
| images | 78.5 | 78.1 | 76.9 | 67.6 | 78.2 | 68.5 | 62.9 | **85.8** | 51.2 | 61.0 | 72.5 | 62.0 | 82.6 | 84.8 | 84.8 |
| OnWN | 78.8 | 79.5 | 75.7 | 67.7 | 78.8 | 76.9 | 61.7 | 85.1 | 23.3 | 58.4 | 75.2 | 61.1 | 82.8 | 84.5 | **88.1** |
| tweet news | 76.4 | 75.8 | 74.2 | 58.0 | 76.9 | 58.7 | 48.2 | **78.7** | 39.9 | 51.2 | 65.1 | 64.7 | 70.1 | 77.5 | 77.5 |
| STS14 | 70.9 | 71.3 | 69.5 | 57.7 | 70.9 | 59.8 | 51.5 | **75.8** | 31.4 | 54.2 | 63.8 | 59.5 | 68.5 | 73.5 | 74.8 |
| answers-forum | 68.3 | 65.1 | 62.6 | 32.8 | 67.4 | 51.9 | 50.7 | **73.1** | 36.1 | 30.5 | 45.6 | 38.8 | 63.9 | 70.1 | 70.7 |
| answers-student | 78.2 | 77.8 | 78.1 | 64.7 | 78.2 | 71.5 | 55.7 | 72.9 | 33.0 | 63.0 | 63.9 | 69.2 | 70.4 | 75.9 | **79.6** |
| belief | 76.2 | 75.4 | 72.0 | 51.9 | 75.9 | 61.7 | 52.6 | **78** | 24.6 | 40.5 | 49.5 | 53.2 | 71.8 | 75.3 | 75.3 |
| headline | 74.8 | 75.2 | 73.5 | 65.3 | 75.1 | 64.0 | 56.6 | **78.6** | 43.6 | 61.8 | 70.9 | 69.0 | 70.7 | 75.9 | 76.8 |
| images | 81.4 | 80.3 | 77.5 | 71.4 | 81.1 | 70.4 | 64.2 | **85.8** | 17.7 | 67.5 | 72.9 | 69.9 | 81.5 | 84.1 | 84.1 |
| STS15 | 75.8 | 74.8 | 72.7 | 57.2 | 75.6 | 63.9 | 56.0 | **77.7** | 31.0 | 52.7 | 60.6 | 60.0 | 71.7 | 76.3 | 77.3 |
| SICK14 | 71.6 | 71.6 | 70.7 | 61.2 | 71.2 | 63.9 | 59.0 | 72.9 | 49.8 | 65.9 | 69.4 | 66.4 | 72.2 | 72.9 | **73.4** |
| Twitter15 | 52.9 | 52.8 | 53.7 | 45.1 | 52.9 | 47.6 | 36.1 | 50.2 | 24.7 | 30.3 | 33.8 | 36.3 | 48.0 | 49.0 | **54.9** |

Table 8: Experimental results (Pearsons r × 100) on textual similarity tasks on STS 16. The highest score in each row is in boldface.

| Tasks | Skip thoughts | LSTM | Tree LSTM | Sent2Vec | Doc2Vec | Glove Avg | Glove tf-idf | PSL Avg | PSL tf-idf | Glove +WR | PSL +WR | P-SIF +PSL |
|---|---|---|---|---|---|---|---|---|---|---|---|---|
| headlines | 51.019 | 75.7 | 74.08 | 75.06 | 69.16 | 49.66 | 52.76 | 70.86 | 72.24 | 72.86 | 74.48 | **75.6** |
| plagiarism | 66.708 | 71.73 | 67.62 | 80.06 | 80.6 | 59.84 | 61.48 | 77.96 | 80.06 | 79.46 | 79.74 | **81.6** |
| post editing | 69.947 | 72.31 | 70.65 | 82.85 | 82.85 | 59.89 | 62.34 | 80.41 | 81.45 | 82.03 | 82.05 | **83.7** |
| answer answer | 28.626 | 44.17 | 52.27 | 57.73 | 41.12 | 19.8 | 22.47 | 38.5 | 41.56 | 58.15 | 59.98 | **60.2** |
| question question | 40.459 | 60.69 | 55.26 | 73.03 | **73.03** | 46.84 | 56.58 | 48.69 | 59.1 | 69.36 | 66.41 | 67.2 |
| STS16 | 51.4 | 64.9 | 64.0 | **73.7** | 69.4 | 47.2 | 51.1 | 63.3 | 66.9 | 72.4 | 72.5 | **73.7** |

Table 9: Results on similarity, entailment, and sentiment tasks. The row for similarity (SICK) shows Pearsons r × 100 and the last two rows show accuracy. The highest score in each row is in boldface. Results in Column 2 to 6 are collected from (Wieting et al., 2016a), and those in Column 7 for skip-thought are from (Kiros et al., 2015), Column 8 for PSL + WR are from (Arora et al., 2017).

| Tasks | PP | DAN | RNN | LSTM (no) | LSTM (o.g.) | skip thought | PSL +WR | P-SIF +PSL |
|---|---|---|---|---|---|---|---|---|
| similarity(SICK) | 84.9 | 85.96 | 73.13 | 85.45 | 83.41 | 85.8 | 86.3 | **87.6** |
| entailment(SICK) | 83.1 | 84.5 | 76.4 | 83.2 | 82.0 | - | 84.6 | **85.5** |
| sentiment(SST) | 79.4 | 83.4 | 86.5 | 86.6 | **89.2** | - | 82.2 | 86.4 |

## C  CLASS PERFORMANCE: 20NEWSGROUP

We also report the precision and recall results of separate 20 classes of the 20 NewsGroup dataset. We compared our embedding (P-SIF) with Bag of Words, SCDV embeddings and our embeddings. Table 10, P-SIF embedding outperform all other embeddings on multiple classes.

Table 10: Class performance on the 20newsgroup dataset. P-SIF represent our embedding. P-SIF (Doc2VecC) represent embedding obtained using Doc2VecC trained word-vectors with 20 clusters.

| Class Name | BoW | | | SCDV | | | P-SIF | | | P-SIF (Doc2VecC) | | |
|---|---|---|---|---|---|---|---|---|---|---|---|---|
| | Pre. | Rec. | F-mes | Pre. | Rec. | F-mes | Pre. | Rec. | F-mes | Pre. | Rec. | F-mes |
| alt.atheism | 67.8 | 72.1 | 69.88 | 80.2 | 79.5 | 79.85 | 83.3 | 80.2 | **81.72** | 83 | 79.9 | 81.42 |
| comp.graphics | 67.1 | 73.5 | 70.15 | 75.3 | 77.4 | 76.34 | 76.6 | 78.1 | 77.34 | 76.8 | 79.2 | **77.98** |
| comp.os.ms-windows.misc | 77.1 | 66.5 | 71.41 | 78.6 | 77.2 | **77.89** | 76.3 | 77.7 | 76.99 | 77.2 | 78.2 | 77.7 |
| comp.sys.ibm.pc.hardware | 62.8 | 72.4 | 67.26 | 75.6 | 73.5 | **74.54** | 73.4 | 74.5 | 73.95 | 71.1 | 74.2 | 72.62 |
| comp.sys.mac.hardware | 77.4 | 78.2 | 77.8 | 83.4 | 85.5 | 84.44 | 87.1 | 84.4 | 85.73 | 87.5 | 87.5 | **87.5** |
| comp.windows.x | 83.2 | 73.2 | 77.88 | 87.6 | 78.6 | 82.86 | 89.3 | 78 | 83.27 | 88.8 | 78.5 | **83.33** |
| misc.forsale | 81.3 | 88.2 | 84.61 | 81.4 | 85.9 | 83.59 | 82.7 | 88 | **85.27** | 82.4 | 86.4 | 84.35 |
| rec.autos | 80.7 | 82.8 | 81.74 | 91.2 | 90.6 | 90.9 | 93 | 90.1 | 91.53 | 92.8 | 90.7 | **91.74** |
| rec.motorcycles | 92.3 | 87.9 | 90.05 | 95.4 | 95.7 | 95.55 | 93.6 | 95.5 | 94.54 | 97 | 96.5 | **96.75** |
| rec.sport.baseball | 89.8 | 89.2 | 89.5 | 93.2 | 94.7 | 93.94 | 93.3 | 95.2 | 94.24 | 95.2 | 95.7 | **95.45** |
| rec.sport.hockey | 93.3 | 93.7 | 93.5 | 96.3 | 99.2 | 97.73 | 95.6 | 98.5 | 97.03 | 96.8 | 98.8 | **97.79** |
| sci.crypt | 92.2 | 86.1 | 89.05 | 92.5 | 94.7 | 93.59 | 89.8 | 93.2 | 91.47 | 93.4 | 96.7 | **95.02** |
| sci.electronics | 70.9 | 73.3 | 72.08 | 74.6 | 74.9 | 74.75 | 79.6 | 78.6 | **79.1** | 78 | 79.3 | 78.64 |
| sci.med | 79.3 | 81.3 | 80.29 | 91.3 | 88.4 | 89.83 | 91.9 | 88.6 | 90.22 | 92.7 | 89.9 | **91.28** |
| sci.space | 90.2 | 88.3 | 89.24 | 88.5 | 93.8 | 91.07 | 89.4 | 94 | 91.64 | 90.7 | 94.4 | **92.51** |
| soc.religion.christian | 77.3 | 87.9 | 82.26 | 83.3 | 92.3 | 87.57 | 84 | 94.3 | 88.85 | 86 | 92.5 | **89.13** |
| talk.politics.guns | 71.7 | 85.7 | 78.08 | 72.7 | 90.6 | 80.67 | 73.1 | 91.2 | 81.15 | 77.3 | 89.8 | **83.08** |
| talk.politics.mideast | 91.7 | 76.9 | 83.65 | 96.2 | 95.4 | 95.8 | 97 | 94.5 | 95.73 | 97.5 | 94.2 | **95.82** |
| talk.politics.misc | 71.7 | 56.5 | 63.2 | 80.9 | 59.7 | 68.7 | 81 | 59 | 68.27 | 82 | 62 | **70.61** |
| talk.religion.misc | 63.2 | 55.4 | 59.04 | 73.5 | 57.2 | 64.33 | 72.2 | 59 | **64.94** | 67.4 | 62.4 | 64.8 |

## D  QUALITATIVE RESULTS

Table 11 represent successful example pair from STS 2012 MSRvid dataset where P-SIF give similarity score closer to ground truth than SIF. Table 12 represent failed example pair from STS 2012 MSRvid dataset where SIF give similarity score closer to ground truth than P-SIF. We now introduce the headline notation use in the Table 11 and 12.

- GT : represent the given ground truth similarity score in range of 0-5.
- NGT : represent normalized ground truth similarity score. NGT is obtain by dividing GT score by 5 so that it's in range of 0-1.
- $SIF_{sc}$ : represent the SIF embedding similarity score in range of 0-1.
- $P\text{-}SIF_{sc}$ : represent the P-SIF embedding similarity score in range of 0-5.
- $SIF_{err}$ : represent absolute error $\|SIF_{sc} - NGT\|$ between normalized ground truth similarity score and SIF embedding similarity score.
- $P\text{-}SIF_{err}$ : represent absolute error $\|P\text{-}SIF_{sc} - NGT\|$ between ground truth similarity score and P-SIF embedding similarity score.
- $Diff_{err}$ : represent absolute difference between $SIF_{err}$ and $P\text{-}SIF_{err}$. Examples where P-SIF perform better $Diff_{err} = P\text{-}SIF_{err} - SIF_{err}$ (used in Table 11). Examples where SIF perform better $Diff_{err} = SIF_{err} - P\text{-}SIF_{err}$ (used in Table 12)
- $Rel_{err}$ : represent relative difference between $SIF_{err}$ and $P\text{-}SIF_{err}$. Example where P-SIF perform better $Rel_{err} = \frac{Diff_{err}}{SIF_{err}}$ (used in Table 11). Examples where SIF perform better $Rel_{err} = \frac{Diff_{err}}{P\text{-}SIF_{err}}$ (used in Table 12)

## E  QUALITATIVE EXAMPLE

Let's consider a corpus ($C$) with $N$ documents with corresponding most frequent words vocabulary ($V$). Figure 3 represents the word-vectors space of $V$, where similar meaning words are closer. We can apply sparse coding and partition the words-vector space in five (total topics $K = 5$) topic vector spaces. Some words are polysemic and belong to multiple topics with some proportion, as shown in Figure 3. For example, words such as *baby*, *person*, *dog* and *kangaroo*, belong to multiple topics with

Table 11: STS 2012 MSRVid example where P-SIF score were closer to the ground truth, whereas SIF score were more further from the ground truth

| sentence1 | sentence2 | GT | NGT | $SIF_{sc}$ | $P\text{-}SIF_{sc}$ | $SIF_{err}$ | $P\text{-}SIF_{err}$ | $Diff_{err}$ | $Rel_{err}$ |
|---|---|---|---|---|---|---|---|---|---|
| People are playing baseball . | The cricket player hit the ball . | 0.5 | 0.1 | 0.2928 | 0.0973 | 0.1928 | 0.0027 | 0.1901 | 0.986 |
| A woman is carrying a boy . | A woman is carrying her baby . | 2.333 | 0.4666 | 0.5743 | 0.4683 | 0.1077 | 0.0017 | 0.106 | 0.9843 |
| A man is riding a motorcycle . | A woman is riding a horse . | 0.75 | 0.15 | 0.5655 | 0.157 | 0.4155 | 0.007 | 0.4085 | 0.9833 |
| A woman slices a lemon . | A man is talking into a microphone . | 0 | 0 | -0.1101 | -0.0027 | 0.1101 | 0.0027 | 0.1074 | 0.9754 |
| A man is hugging someone . | A man is taking a picture . | 0.4 | 0.08 | 0.2021 | 0.0767 | 0.1221 | 0.0033 | 0.1188 | 0.9731 |
| A woman is dancing . | A woman plays the clarinet . | 0.8 | 0.16 | 0.3539 | 0.1653 | 0.1939 | 0.0053 | 0.1886 | 0.9727 |
| A train is moving . | A man is doing yoga . | 0 | 0 | 0.1674 | -0.0051 | 0.1674 | 0.0051 | 0.1623 | 0.9695 |
| Runners race around a track . | Runners compete in a race . | 3.2 | 0.64 | 0.7653 | 0.6438 | 0.1253 | 0.0038 | 0.1214 | 0.9694 |
| A man is driving a car . | A man is riding a horse . | 1.2 | 0.24 | 0.3584 | 0.2443 | 0.1184 | 0.0043 | 0.114 | 0.9636 |
| A man is playing a guitar . | A woman is riding a horse . | 0.5 | 0.1 | -0.0208 | 0.0955 | 0.1208 | 0.0045 | 0.1163 | 0.9629 |
| A man is riding on a horse . | A girl is riding a horse . | 2.6 | 0.52 | 0.6933 | 0.5082 | 0.1733 | 0.0118 | 0.1615 | 0.9319 |
| A woman is deboning a fish . | A man catches a fish . | 1.25 | 0.25 | 0.4538 | 0.2336 | 0.2038 | 0.0164 | 0.1875 | 0.9196 |
| A man is playing a guitar . | A man is eating pasta . | 0.533 | 0.1066 | -0.0158 | 0.0962 | 0.1224 | 0.0104 | 0.112 | 0.915 |
| A woman is dancing . | A man is eating . | 0.143 | 0.0286 | -0.1001 | 0.0412 | 0.1287 | 0.0126 | 0.1161 | 0.9023 |
| The ballerina is dancing . | A man is dancing . | 1.75 | 0.35 | 0.512 | 0.3317 | 0.162 | 0.0183 | 0.1437 | 0.8871 |
| A woman plays the guitar . | A man sings and plays the guitar . | 1.75 | 0.35 | 0.5036 | 0.3683 | 0.1536 | 0.0183 | 0.1353 | 0.8807 |
| A girl is styling her hair . | A girl is brushing her hair . | 2.5 | 0.5 | 0.7192 | 0.5303 | 0.2192 | 0.0303 | 0.1889 | 0.8618 |
| A guy is playing hackysack | A man is playing a key-board . | 1 | 0.2 | 0.3718 | 0.2268 | 0.1718 | 0.0268 | 0.145 | 0.8441 |
| A man is riding a bicycle . | A monkey is riding a bike . | 2 | 0.4 | 0.6891 | 0.4614 | 0.2891 | 0.0614 | 0.2277 | 0.7876 |
| A woman is swimming underwater . | A man is slicing some carrots . | 0 | 0 | -0.2158 | -0.0562 | 0.2158 | 0.0562 | 0.1596 | 0.7397 |
| A plane is landing . | A animated airplane is landing . | 2.8 | 0.56 | 0.801 | 0.6338 | 0.241 | 0.0738 | 0.1672 | 0.6937 |
| The missile exploded . | A rocket exploded . | 3.2 | 0.64 | 0.8157 | 0.6961 | 0.1757 | 0.0561 | 0.1196 | 0.6806 |
| A woman is peeling a potato . | A woman is peeling an apple . | 2 | 0.4 | 0.6938 | 0.5482 | 0.2938 | 0.1482 | 0.1456 | 0.4956 |
| A woman is writing . | A woman is swimming . | 0.5 | 0.1 | 0.3595 | 0.2334 | 0.2595 | 0.1334 | 0.1261 | 0.4859 |
| A man is riding a bike . | A man is riding on a horse . | 2 | 0.4 | 0.6781 | 0.564 | 0.2781 | 0.164 | 0.1142 | 0.4105 |
| A panda is climbing . | A man is climbing a rope . | 1.6 | 0.32 | 0.4274 | 0.3131 | 0.1074 | 0.0069 | 0.1005 | 0.9361 |
| A man is shooting a gun . | A man is spitting . | 0 | 0 | 0.2348 | 0.1305 | 0.2348 | 0.1305 | 0.1043 | 0.444 |

Table 12: STS 2012 MSRVid example where P-SIF score were far away from ground truth, whereas SIF score were closer to actual ground truth

| sentence1 | sentence2 | GT | NGT | $SIF_{sc}$ | $P\text{-}SIF_{sc}$ | $SIF_{err}$ | $P\text{-}SIF_{err}$ | $Diff_{err}$ | $Rel_{err}$ |
|---|---|---|---|---|---|---|---|---|---|
| takes off his sunglasses . | A boy is screaming . | 0.5 | 0.1 | 0.1971 | 0.3944 | 0.0971 | 0.2944 | 0.1973 | 0.6703 |
| The rhino grazed on the grass . | A rhino is grazing in a field . | 4 | 0.8 | 0.7275 | 0.538 | 0.0725 | 0.262 | 0.1895 | 0.7234 |
| An animal is biting a persons finger . | A slow loris is biting a persons finger . | 3 | 0.6 | 0.6018 | 0.7702 | 0.0018 | 0.1702 | 0.1684 | 0.9892 |
| Animals are playing in water . | Two men are playing ping pong . | 0 | 0 | 0.0706 | 0.2238 | 0.0706 | 0.2238 | 0.1532 | 0.6846 |
| Someone is feeding a animal . | Someone is playing a piano . | 0 | 0 | -0.0037 | 0.1546 | 0.0037 | 0.1546 | 0.1509 | 0.976 |
| The lady sliced a tomatoe . | Someone is cutting a tomato . | 4 | 0.8 | 0.693 | 0.5591 | 0.107 | 0.2409 | 0.1339 | 0.5559 |
| The lady peeled the potatoe . | A woman is peeling a potato . | 4.75 | 0.95 | 0.7167 | 0.5925 | 0.2333 | 0.3575 | 0.1242 | 0.3474 |
| A man is slicing something . | A man is slicing a bun . | 3 | 0.6 | 0.5976 | 0.4814 | 0.0024 | 0.1186 | 0.1162 | 0.9802 |
| A boy is crawling into a dog house . | A boy is playing a wooden flute . | 0.75 | 0.15 | 0.1481 | 0.2674 | 0.0019 | 0.1174 | 0.1155 | 0.9839 |
| A man and woman are talking . | A man and woman is eating . | 1.6 | 0.32 | 0.3574 | 0.4711 | 0.0374 | 0.1511 | 0.1137 | 0.7527 |
| A man is cutting a potato . | A woman plays an electric guitar . | 0.083 | 0.0166 | -0.1007 | -0.2128 | 0.1173 | 0.2294 | 0.112 | 0.4884 |
| A person is cutting a meat . | A person riding a mechanical bull | 0 | 0 | 0.0152 | 0.1242 | 0.0152 | 0.1242 | 0.1091 | 0.8778 |
| A woman is playing the flute . | A man is playing the guitar . | 1 | 0.2 | 0.1942 | 0.0876 | 0.0058 | 0.1124 | 0.1065 | 0.948 |

significant proportion. Words and corresponding vectors in these topic vector spaces are represented by topic numbers in subscript. Table 13 shows an example pair from the STS Task 2012 MSRVid dataset, and the corresponding SIF (averaging) and P-SIF (partition averaging) representation vectors. We can see that in SIF representation we are averaging words vectors which semantically have different meanings. The document is represented in the same $d$ dimensional word-vectors space. Overall, the SIF represent the document as a single point in the vector space and doesn't take account of different semantic meanings of the topics. Whereas, in P-SIF representation we treat the 5 different semantic topics distinctly. Words belonging to different semantic topics are separated by concatenation ($\oplus$) as they represent different meanings, whereas words coming from same topic are average as it's represent same meaning. The final document vector $\vec{v}_{d_n}$ has more representational power as it's represented in a higher $5 \times d$ dimensional vector space. Thus partitioned averaging with topic weighting is important for representing documents. Empirically, P-SIF gave score dissimilar sentences $(d_n^1, d_n^2)$ 0.16 (0-1 scale) where the ground truth of 0.15 (rescale to 0-1 scale), whereas SIF gave similarity score of 0.57 (0-1 scale), farther then ground score. Thus we obtain a relative improvement of 98% in the error difference from ground truth. Here, simple averaging-based embedding of $d_n^1$ and $d_n^2$, bring the document representation closer. But, partitioned based averaging P-SIF of $d_n^1$ and $d_n^2$ as, project the document in a higher dimensional space.

Table 13: STS Task 2012 MSRVid dataset similarity example pair where P-SIF gave score dissimilar sentences $(d_n^1, d_n^1)$ 0.16 (0-1 scale) where the ground truth of 0.15 (rescale to 0-1 scale), whereas SIF gave similarity score of 0.57 (0-1 scale). Thus, we obtain a relative improvement of 98% in the error difference. Here, $\oplus$ represent concatenation. $\vec{v}_{\text{zero}}$ is the zero padding vector.

| | Document 1 $(d_n^1)$ | Document 2 $(d_n^2)$ | Score |
|---|---|---|---|
| Doc | A man is riding a motorcycle | A woman is riding a horse | 0.15 |
| SIF | $\vec{v}_{\text{man}_2} + \vec{v}_{\text{riding}_3} + \vec{v}_{\text{motorcycle}_4}$ | $\vec{v}_{\text{woman}_1} + \vec{v}_{\text{riding}_3} + \vec{v}_{\text{horse}_5}$ | 0.57 |
| P-SIF | $\vec{v}_{\text{zero}_1} \oplus \vec{v}_{\text{man}_2} \oplus \vec{v}_{\text{riding}_3} \oplus \vec{v}_{\text{motorcycle}_4} \oplus \vec{v}_{\text{zero}_5}$ | $\vec{v}_{\text{women}_1} \oplus \vec{v}_{\text{zero}_2} \oplus \vec{v}_{\text{riding}_3} \oplus \vec{v}_{\text{zero}_4} \oplus \vec{v}_{\text{horse}_5}$ | 0.16 |

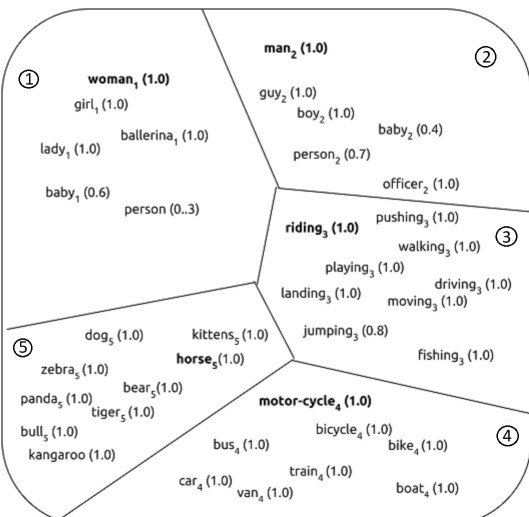

Figure 3: Word vector space for corpus C. Words in different topic is represented by different subscript and separated by hyperplanes. Bold represent words from example documents.

## F  EXPERIMENTAL DETAILS

### F.1  TEXTUAL SIMILARITY TASK:

We use the PARAGRAM-SL999 (PSL) from (Wieting et al., 2015) as word embeddings, obtained by training on PPDB (Ganitkevitch et al., 2013) dataset [9]. We use the fix weighting parameter $\alpha$ value of $10^{-3}$, and the word frequencies $p(w)$ are estimated from the common-crawl dataset. We tune the number of contexts $(K)$ to minimize the reconstruction loss over word vectors. We fix the non-zero coefficient $m = K/2$, for the SIF experiments. For GMM based partitioning of the words vocabulary we tune the number of cluster parameter $K$ through 5-fold cross validation.

### F.2  TEXTUAL CLASSIFICATION TASK:

We fix the document embeddings and only learn the classifier. We learn word vector embedding using Skip-Gram with a window size of 10, Negative Sampling (SGNS) of 10 and minimum word frequency of 20. We use 5-fold cross-validation on the $F1$ score to tune hyperparameters. We use LinearSVM for multi-class classification and Logistic regression with the OneVsRest setting for multi-label classification. We fix the number of dictionary elements either 40 or 20 (with Doc2vecC initialize word vectors) and non-zero coefficient to $m = K/2$ during dictionary learning for all experiments. We use the best parameter settings as reported in all our baselines to generate their results. We use 200 dimensions for tf-idf weighted word-vector model, 400 for paragraph vector model, 80 topics and 400 dimensional vectors for TWE, NTSG, LTSG and 60 topics and 200 dimensional word vectors for SCDV (Mekala et al., 2017). We will released P-SIF's embedding source code with necessary parameters details and other data-sets used in the paper for reproducing results.

---

[9] For a fair comparison with SIF we uses PSL vectors instead of unsupervised Glove and Word2Vec vectors

# G  CODE FLOW ARCHITECTURE

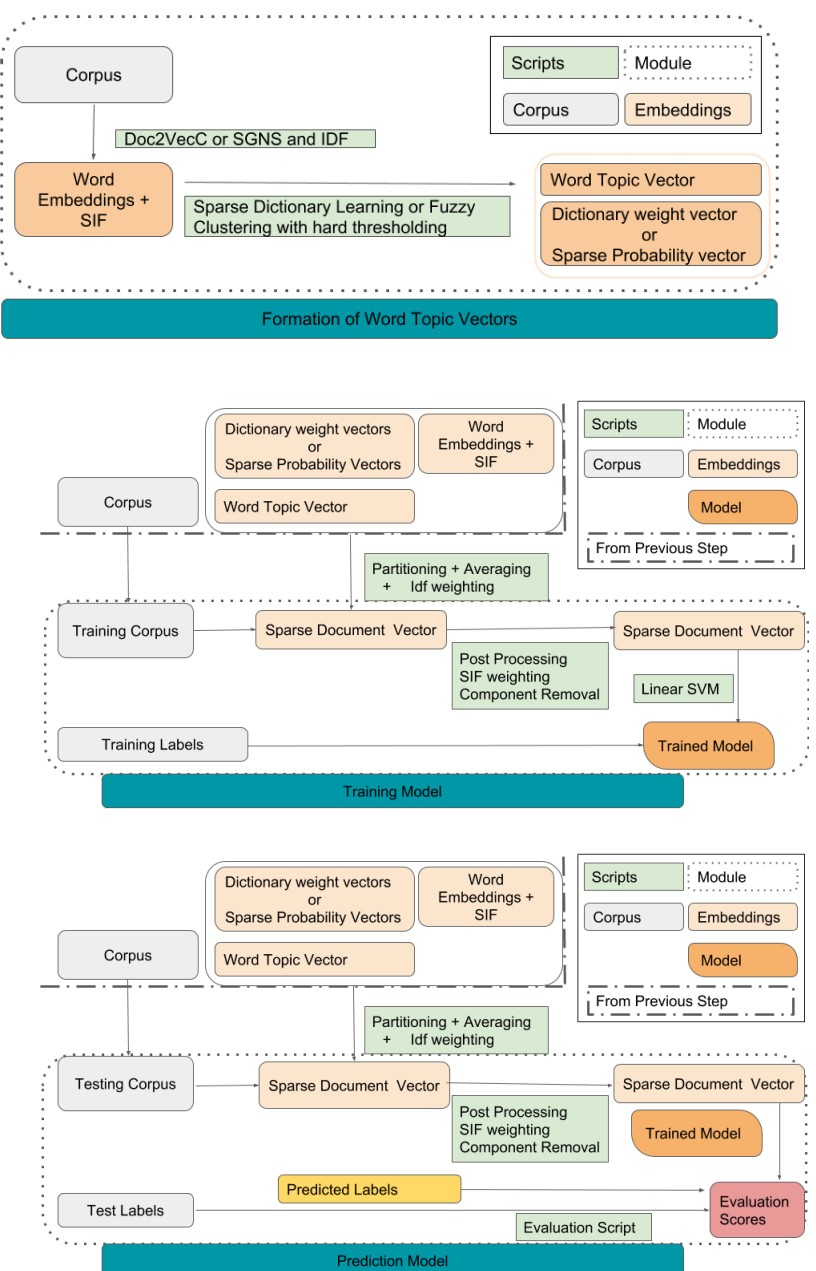

# H  HIGH LEVEL FLOW

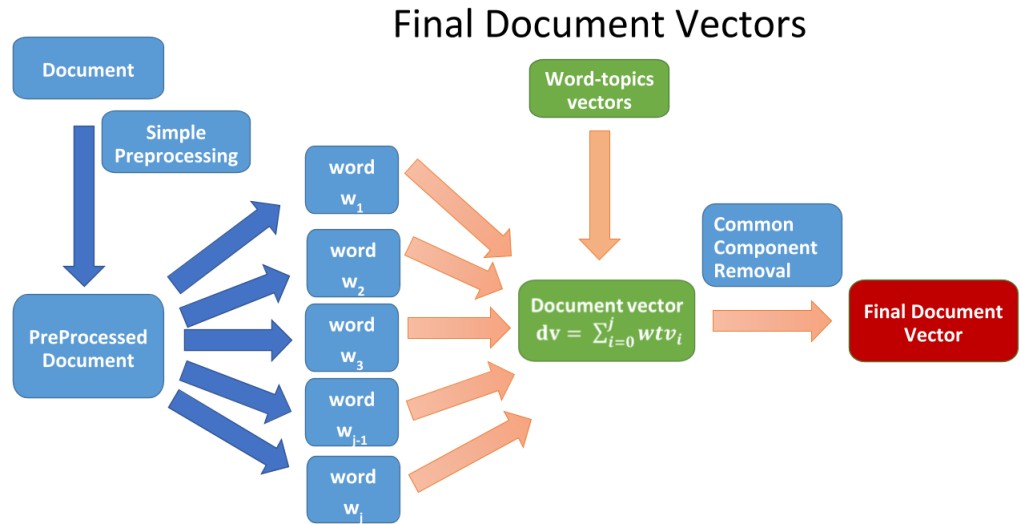

## I   PROOF SKETCH: DERIVATION OF OUR EMBEDDING

To derive our embedding, we propose a generative model which treats corpus generation as a dynamic process, where the $t^{th}$ word is produced at step $t$. The process is driven by random walk over a unit norm sphere with centre at the origin. Let, $\vec{v}_{c_t}$ be the vector from origin to current walk point at time $t$. We called this vector as context vector, as it represents the context in the discussion. Below, we introduce formal notation needed for the discussion

- $C$ represents text corpus and $V$ represents vocabulary of words in corpus.
- $\vec{v}_w \in R^d$ represents the word vector of word w , where d is dimension of word vector.
- $c_t$ represents context and $\vec{v}_{c_t} \in R^d$ represents context vector at time $t$, where d is dimension of context vector.
- $Z_c$ represents partition function for the random context vector $\vec{v}_{c_t}$, given by $Z_c = \sum_w \exp(\langle \vec{v}_{c_t}, \vec{v}_w \rangle)$.
- $p(w)$ represents the unigram probability of word $w$ in the corpus.
- $c_0$ and $\vec{v}_{c_0} \in R^d$ represent common context and its corresponding context vector based on syntax.

Using log linear model of Mnih & Hinton (2007), we define the probability of observing a word $w$ from the random walk with current context $c_t$ at time $t$ as

$$Pr[w|c_t] \propto \exp(\langle \vec{c}_t, \vec{v}_w \rangle) \tag{3}$$

It is easy to show that such random walk under some reasonable assumptions (Arora et al. (2016a)) can gives word-word co-occurrence probabilities similar to empirical works like word2vec (Mikolov et al. (2013a)) and Glove (Pennington et al. (2014a)). To account for frequent stop-words which occur more often regardless of context and common context related to document syntax, two correction terms need be added: one based on $p(w)$ and other on common context vector $\vec{v}_{c_0}$ in Equation equation 3. These terms allow words with low inner product with $\vec{c}_t$, a chance to appear either from term $p(w)$ if they are frequent or by common context $\vec{c}_0$ , if they have large dot product with $\vec{c}_0$. Given a context vector $c_t$, the probability of a word $w$ in document $d$ being generated by context $c_t$ is given by,

$$Pr[w|c_t] = \lambda p(w) + (1 - \lambda) \frac{\exp(\langle \vec{c}_t', \vec{v}_w \rangle)}{Z_{c_t'}} \tag{4}$$

where, $\vec{c_t'} = \beta \vec{c}_0 + (1 - \beta)\, \vec{c}_t$, $\langle \vec{c}_0, \vec{c}_t \rangle = 0$, $\lambda$ and $\beta$ are scalar hyper-parameters.

For generating a document from above random walk based latent variable model, we consider the following two assumptions :

1. Total number of theme/topics in the entire corpus is $K$. The $K$ themes/topics can be determine by sparse dictionary learning as shown by Arora et al. (2016b) over word vectors $\vec{v}_w = \sum_{j=1}^{m} \alpha_{w,j} \vec{A}_j + \vec{\eta}_w$, where, $\vec{A}_j$'s are unit norm vectors representing basis of theme, $\alpha_{w,j}$ are coefficient determining whether $w$ is generated with $\vec{A}_j$ [10] , and $\vec{\eta}_w$ is a noise vector.
2. Word vectors $\vec{v}_w$ are uniformly distributed, thus making the partition function $Z_c$, roughly same in all direction for given context $c$, where $c$ belongs to one of the $K$ themes/topics, as described earlier. The context vector does not change significantly much while words are generated from random walk, as shown by Arora et al. (2017) except during jumps a.k.a when theme/topic change.

For a document $d$, the likelihood of document is being generated by the $K$ contexts, is given by :

---

[10]   In practice, only $k$ (much lesser than total number of themes/topic $K$) $\alpha_{w,j}$ will be non-zero, because a word can't belong to all contexts (exception for frequent stop-words)

$$p(d|\{c_1, c_2 \ldots c_K\}) \propto \prod_{j=1}^{K} \prod_{\{w \in d\}} p(w|c_j) \tag{5}$$

$$= \prod_{j=1}^{K} \prod_{w \in d} \left[ \lambda p(w) + (1-\lambda) \frac{\exp(\langle \vec{v}_w, \vec{v}_{c_j} \rangle)}{Z_j} \right] \tag{6}$$

Let,

$$f_w(c_j) = \log \left[ \lambda p(w) + (1-\lambda) \frac{\exp(\langle \vec{v}_w, \vec{v}_{c_j} \rangle)}{Z_j} \right] \tag{7}$$

Here, $p(w|c_j)$ is the probability that word $w$ is generated by context $c_j$, the value of which is determine by 1) overall frequency of word $w$ in corpus a.k.a prior probability ($p(w)$) and 2) relative frequency of $w$ appear with context $j$ w.r.t other contexts (determine by $\alpha_{(w,j)}$).

Using simple calculus, treating $p(w)$ as constant, we can show that $\nabla(f_w(c_j))$ equals,

$$\frac{1}{\lambda p(w) + (1-\lambda) \exp(\langle \vec{v}_w, \vec{v}_{c_j} \rangle)/Z_j} * \frac{1-\lambda}{Z_j} \exp(\langle \vec{v}_w, \vec{v}_{c_j} \rangle) \vec{v}_w \tag{8}$$

Then, by using the Taylor expansion, we can show

$$f_w(c_j) \approx f_w(c_j = 0) + \nabla(f_w(c_j = 0))^T \vec{v}_{c_j} \tag{9}$$

$$f_w(c_j) \approx constant + \nabla(f_w(c_j = 0))^T \vec{v}_{c_j} \tag{10}$$

Therefore, the maximum likelihood estimator (MLE) for $\vec{v}_{c_j}$ on the unit sphere (ignoring normalization) is approximately, [11]

$$\arg\max \sum_{w \in d} f_w(c_j) \propto \sum_{w \in d} \frac{a}{p(w) + a} \vec{v}_w \tag{11}$$

here, $a = \frac{1-\lambda}{\lambda Z_j}$

Thus, the MLE estimate is approximately a weighted average of the word vectors generated from context $j$ in the document $d$ from random walk. We can get the overall context representation $\vec{v}_{c_d}$ of document, by simple concatenation over all $K$ themes/topics.

$$\vec{v}_{c_d} = \bigoplus_{j=1}^{K} \vec{v}_{c_j} \tag{12}$$

Here, $\bigoplus$ represents concatenation operation. For a document if no words is generated from the context $c_j$ the we can substitute the context vector $\vec{v}_{c_j}$ by $\vec{0}$ vector, for representation $\vec{v}_{c_d}$ in $K \times d$ dimensions.

**Relation to SIF model:** Arora et al. (2017), shows under the two assumptions :

- uniform distribution of word vectors $\vec{v}_w$'s which implies that the partition function $Z_t$ is roughly same in all direction for the a sentence.
- the context vector $\vec{v}_{c_t}$, remain constant while the words in the sentence are emitted, implying replacing of $\vec{v}_{c_t}$ in sentence's by a single vector $\vec{v}_{c_s}$ and partition function by $Z_s$

---

[11] Note that $argmax_{c:\|\vec{c}\|=1} C + \langle \vec{c}, \vec{g} \rangle = \frac{\vec{g}}{\|\vec{g}\|}$ for any constant C

the sentence embedding of a sentence can be obtained by $\vec{v}_{c_s} = \sum_{\{w \in s\}} \frac{a}{p(w)+a} \vec{v}_w$, here, $a = \frac{1-\lambda}{\lambda Z_s}$.

However, the above assumptions does not hold true for a document with multiple sentences, where one can expect to have more frequent random jumps during random walk [12]. Instead of assuming a single context for whole document $c_h$, we assume that the total number of theme/topics over a given corpus is bounded by $K$ (as shown by Arora et al. (2016b)) and the random walk can perform jumps to switch context from one context to the rest $K - 1$ contexts. The partition function remain same in all directions for only words coming from a same context $c_j$, instead of words coming from all $K$ contexts. Thus, our approach is generalization of sentence embedding approach by Arora et al. (2017) (special case K = 1).

## J    KERNEL CONNECTION TO WORD MOVER DISTANCE

Below, we introduce formal notation needed for the discussion:

- $C$ represents text corpus and $V$ represents vocabulary of words in corpus.
- $\vec{wv}_w \in R^d$ represents the word vector of word w , where d is dimension of word vector.
- $\vec{tv}_w \in R^K$ represents the theme/topic vector of word w, where K is number of themes/topics.

$$t_{w,j} = \alpha_{w,j} = P(j|w)$$

- $d_A$, $d_B$ represent two document containing $n$ and $m$ words respectively. $w_1^A, w_2^A \ldots w_n^A$ represent words of $d_A$ and $w_1^B, w_2^B \ldots w_n^B$ represents word of $d_B$.

Consider the following document similarity kernels:

$$K^1(d_A, d_B) = \frac{1}{nm} \sum_{i=1}^{n} \sum_{j=1}^{m} \langle \vec{wv}_{w_i^A} \cdot \vec{wv}_{w_j^B} \rangle = \mathbb{E}_{i,j} \langle \vec{wv}_{w_i^A} \cdot \vec{wv}_{w_j^B} \rangle$$

$$K^2(d_A, d_B) = \frac{1}{nm} \sum_{i=1}^{n} \sum_{j=1}^{m} \langle \vec{wv}_{w_i^A} \cdot \vec{wv}_{w_j^B} \rangle \times \langle \vec{tv}_{w_i^A} \cdot \vec{tv}_{w_j^B} \rangle = \mathbb{E}_{i,j} \langle \vec{wv}_{w_i^A} \cdot \vec{wv}_{w_j^B} \rangle \times \langle \vec{tv}_{w_i^A} \cdot \vec{tv}_{w_j^B} \rangle$$

$$K^3(d_A, d_B) = \frac{1}{n} \sum_{i=1}^{n} \max_j \vec{wv}_{w_i^A} \cdot \vec{wv}_{w_j^B} \rangle = \mathbb{E}_i \left( \max_j \langle \vec{wv}_{w_i^A} \cdot \vec{wv}_{w_j^B} \rangle \right)$$

$$K^4(d_A, d_B) = \frac{1}{m} \sum_{j=1}^{m} \max_i \langle \vec{wv}_{w_i^A} \cdot \vec{wv}_{w_j^B} \rangle = \mathbb{E}_i \left( \max_i \langle \vec{wv}_{w_i^A} \cdot \vec{wv}_{w_j^B} \rangle \right)$$

$$K^5(d_A, d_B) = K^3(d_A, d_B) + K^4(d_A, d_B)$$

We can conclude the following from the respective kernels:

- $K^1(d_A, d_B)$ represent document similarity between document represented by average word vectors $d_x = \sum_i \vec{wv}_i^x$
- $K^1(d_A, d_B)$ represent document similarity between document represented by partition average word vectors
- $K^3(d_A, d_B)$ represent document similarity between document represented by relax word mover distance when words of $d_A$ are matched to $d_B$
- $K^4(d_A, d_B)$ represent document similarity between document represented by relax word mover distance when words of $d_B$ are matched to $d_A$
- $K^5(d_A, d_B)$ represent document similarity between document represented by word mover distance.

We empirically showed that our proposed embedding (kernel) outperform the word mover distance(Kusner et al., 2015) and word mover embedding (Wu et al., 2018) and many other baselines in Table 14 on several datasets [13].

---

[12] It is trivial to assume that these jumps occur more frequently in multiple sentences document, because of increased chances of context change    [13] For datasets and baseline details refer to (Wu et al., 2018)

Table 14: Comparison of our embedding P-SIF (SGNS) with recently proposed word mover distance and word mover embedding. All dataset and baseline are taken from (Wu et al., 2018)

| Dataset | BB | Twitter | Ohsu umed | CLASSIC | Reu ters | Amazon | 20News Group | RECIPE-L |
|---|---|---|---|---|---|---|---|---|
| BOW | $79.4 \pm 1.2$ | $56.4 \pm 0.4$ | 38.9 | $64.0 \pm 0.5$ | 86.1 | $71.5 \pm 0.5$ | 42.2 | - |
| TF-IDF | $78.5 \pm 2.8$ | $66.8 \pm 0.9$ | 37.3 | $65.0 \pm 1.8$ | 70.9 | $58.5 \pm 1.2$ | 45.6 | - |
| BM25 | 83.1F1.5 | $57.3 \pm 7.8$ | 33.8 | $59.4 \pm 2.7$ | 67.2 | $41.2 \pm 2.6$ | 44.1 | - |
| LSI | $95.7 \pm 0.6$ | $68.3 \pm 0.7$ | 55.8 | $93.3 \pm 0.4$ | 93.7 | $90.7 \pm 0.4$ | 71.1 | - |
| LDA | $93.6 \pm 0.7$ | $66.2 \pm 0.7$ | 49 | $95.0 \pm 0.3$ | 93.1 | $88.2 \pm 0.6$ | 68.5 | - |
| mSDA | $91.6 \pm 0.8$ | $67.7 \pm 0.7$ | 50.7 | $93.1 \pm 0.4$ | 91.9 | $82.9 \pm 0.4$ | 60.5 | - |
| SIF(GloVe) | $97.3 \pm 1.2$ | $57.8 \pm 2.5$ | 67.1 | $92.7 \pm 0.9$ | 87.6 | $94.1 \pm 0.2$ | 72.3 | $71.1 \pm 0.5$ |
| Word2Vec +nbow | $97.3 \pm 0.9$ | $72.0 \pm 1.5$ | 63 | $95.2 \pm 0.4$ | 96.9 | $94.0 \pm 0.5$ | 71.7 | $74.9 \pm 0.5$ |
| Word2Vec +tf-idf | $96.9 \pm 1.1$ | $71.9 \pm 0.7$ | 60.6 | $93.9 \pm 0.4$ | 95.9 | $92.2 \pm 0.4$ | 70.2 | $73.1 \pm 0.6$ |
| PV-DBOW | $97.2 \pm 0.7$ | $67.8 \pm 0.4$ | 55.9 | $97.0 \pm 0.3$ | 96.3 | $89.2 \pm 0.3$ | 71 | $73.1 \pm 0.5$ |
| PV-DM | $97.9 \pm 1.3$ | $67.3 \pm 0.3$ | 59.8 | $96.5 \pm 0.7$ | 94.9 | $88.6 \pm 0.4$ | 74 | $71.1 \pm 0.4$ |
| Doc2VecC | $90.5 \pm 1.7$ | $71.0 \pm 0.4$ | 63.4 | $96.6 \pm 0.4$ | 96.5 | $91.2 \pm 0.5$ | 78.2 | $76.1 \pm 0.4$ |
| Doc2VecC (Train) | $89.2 \pm 1.4$ | $69.8 \pm 0.9$ | 59.6 | $96.2 \pm 0.5$ | 96 | $89.5 \pm 0.4$ | 72.9 | $75.6 \pm 0.4$ |
| KNN-WMD | $95.4 \pm 1.2$ | $71.3 \pm 0.6$ | 55.5 | $97.2 \pm 0.1$ | 96.5 | $92.6 \pm 0.3$ | 73.2 | $71.4 \pm 0.5$ |
| WME(SR) | $95.5 \pm 0.7$ | $72.5 \pm 0.5$ | 55.8 | $96.6 \pm 0.2$ | 96 | $92.7 \pm 0.3$ | 72.9 | $72.5 \pm 0.4$ |
| WME(LR) | $98.2 \pm 0.6$ | $74.5 \pm 0.5$ | 64.5 | $97.1 \pm 0.4$ | 97.2 | $94.3 \pm 0.4$ | 78.3 | $79.2 \pm 0.3$ |
| P-SIF | $\mathbf{99.05 \pm 0.9}$ | $\mathbf{74.39 \pm 0.9}$ | $\mathbf{66.2}$ | $\mathbf{97.95 \pm 0.5}$ | $\mathbf{97.5}$ | $\mathbf{95.17 \pm 0.3}$ | $\mathbf{79.15}$ | $\mathbf{79.86 \pm 0.3}$ |

## K    RECENT BASELINE FOR TEXTUAL SIMILARITY TASK

We compared out P-SIF embedding with many other recently proposed baseline like ELMO (Peters et al., 2018), p-means (Rücklé et al., 2018), FastText (Joulin et al., 2016), Skip-Thoughts (Kiros et al., 2015), (Conneau et al., 2017), Charphrase (Wieting et al., 2016b), WME (Wu et al., 2018) and u-SIF Ethayarajh (2018). We used the SentEval package (Conneau & Kiela, 2018) and embedding evaluation paper Perone et al. (2018) for baselines. Except u-SIF we outperform all other embedding. Our P-SIF results was very close to u-SIF in most tasks. Details results are provided in Table 15

Table 15: Comparison of our P-SIF embedding with recently proposed embedding techniques on various STS tasks. Many baselines taken from (Conneau & Kiela, 2018),Perone et al. (2018). (Wu et al., 2018) and Ethayarajh (2018)

| Task | ELMO orig +all | ELMO orig +top | p-mean | Fast Text | Skip Thou ghts | Infer Sent | Char pharse | WME +PSL | PSIF +PSL | u-SIF +PSL |
|---|---|---|---|---|---|---|---|---|---|---|
| STS 12 | 55 | 54 | 54 | 58 | 41 | 61 | 66 | 62.8 | 65.7 | **65.8** |
| STS 13 | 51 | 49 | 52 | 58 | 29 | 56 | 57 | 56.3 | 63.98 | **65.2** |
| STS 14 | 63 | 62 | 63 | 65 | 40 | 68 | 74.7 | 68.0 | 74.8 | **75.9** |
| STS 15 | 69 | 67 | 66 | 68 | 46 | 71 | 76.1 | 64.2 | 77.29 | 77.6 |
| STS 16 | 64 | 63 | 67 | 64 | 52 | 77 | - | - | **73.7** | 72.3 |
| Average | 60.4 | 59 | 60.4 | 62.6 | 41.6 | 66.6 | 68.45 | 62.85 | 71.09 | **71.36** |

## L    MORE EXPERIMENTAL RESULTS

We also compared our embedding extensively on random hashing of words (instead of GMM) (refer Table 16), p-means (Rücklé et al., 2018) (refer Table 17), and ELMO (Peters et al., 2018) (refer Table 18) embedding on 20NewsGroup. P-SIF outperform all three embedding on 20NewsGroup.

Table 16: Performance on 20NewsGroup classification by uniform random hashing of word-vectors into 40 bins/groups

| Run | Accuracy | Precision | Recall | F1-meas |
|---|---|---|---|---|
| 1 | 83.86 | 83.92 | 83.86 | 83.71 |
| 2 | 83.68 | 83.81 | 83.68 | 83.54 |
| 3 | 84.17 | 84.25 | 84.17 | 84.04 |
| Average | 83.9 | 83.99 | 83.9 | 83.76 |
| P-SIF (Doc2VecC) | **86.0** | **86.1** | **86.1** | **86.0** |
| P-SIF | **85.4** | **85.5** | **85.4** | **85.2** |

Table 17: Performance on 20NewsGroup classification using p-means embedding

| Power | z-norm | Dimension | Accuracy | Precision | Recall | F1-score |
|---|---|---|---|---|---|---|
| 1,2 | FALSE | 400 | 81.97 | 81.91 | 81.97 | 81.57 |
| 1 | FALSE | 200 | 81.55 | 81.48 | 81.55 | 81.06 |
| 1,2,3 | FALSE | 600 | 80.63 | 80.48 | 80.63 | 80.34 |
| 1,2,-inf | FALSE | 600 | 80.32 | 80.2 | 80.32 | 80.02 |
| 1,2,+inf | FALSE | 600 | 80.01 | 79.91 | 80.01 | 79.67 |
| 1,+inf | FALSE | 400 | 79.59 | 79.46 | 79.59 | 79.25 |
| 1,-inf,+inf | FALSE | 600 | 79.1 | 78.91 | 79.1 | 78.74 |
| 1 | TRUE | 200 | 78.7 | 78.63 | 78.7 | 78.48 |
| 1,2 | TRUE | 400 | 75.23 | 75.15 | 75.23 | 75.03 |
| 1,+inf | TRUE | 400 | 74.46 | 74.17 | 74.46 | 74.19 |
| 1,2,3 | TRUE | 600 | 73.42 | 73.35 | 73.42 | 73.27 |
| 1,2,-inf | TRUE | 600 | 73.25 | 73.15 | 73.25 | 73.08 |
| 1,2,+inf | TRUE | 600 | 72.66 | 72.54 | 72.66 | 72.47 |
| 1,2,3,-inf | TRUE | 800 | 72.28 | 72.32 | 72.28 | 72.18 |
| 1,2,3,-inf,+inf | TRUE | 1000 | 72.15 | 72.33 | 72.15 | 72.11 |
| 1,2,3,+inf | TRUE | 800 | 71.63 | 71.76 | 71.63 | 71.58 |
| 1,2,-inf,+inf | TRUE | 800 | 71.59 | 71.67 | 71.59 | 71.48 |
| 1,-inf,+inf | TRUE | 600 | 71.1 | 70.97 | 71.1 | 70.96 |
| 1,-inf,+inf | TRUE | 600 | 69.21 | 68.79 | 69.21 | 68.67 |
| 1,2,-inf,+inf | FALSE | 800 | 69.16 | 68.73 | 69.16 | 68.61 |
| 1,-inf | FALSE | 400 | 59.6 | 59.04 | 59.6 | 58.63 |
| 1,-inf | TRUE | 400 | 59.59 | 59.02 | 59.59 | 58.61 |
| P-SIF (Doc2VecC) | - | - | **86.0** | **86.1** | **86.1** | **86.0** |
| P-SIF | - | - | **85.4** | **85.5** | **85.4** | **85.2** |

We also compared our embedding extensively on p-means (Rücklé et al., 2018) embedding on Reuters. P-SIF outperform p-means on Reuters for various values of p (refer Table 19).

Table 18: Performance on 20NewsGroup classification using ELMO embedding

| Average | tfidf | layers | dimension | Accuracy | Precision | Recall | f1-score |
|---|---|---|---|---|---|---|---|
| word | FALSE | 1,2 | 2048 | 74.07 | 74.02 | 74.07 | 73.94 |
| word | FALSE | 1,2,3 | 3072 | 73.21 | 73.25 | 73.21 | 73.14 |
| word | FALSE | 2,3 | 2048 | 71.14 | 71.03 | 71.14 | 70.99 |
| word | TRUE | 1,2,3 | 3072 | 70.5 | 70.81 | 70.5 | 70.6 |
| word | FALSE | 2 | 1024 | 70.76 | 70.48 | 70.76 | 70.48 |
| word | FALSE | 3 | 1024 | 70.01 | 69.83 | 70.01 | 69.79 |
| word | FALSE | 1 | 1024 | 69.16 | 68.88 | 69.16 | 68.85 |
| word | TRUE | 1,2 | 2048 | 67.79 | 67.99 | 67.79 | 67.83 |
| word | TRUE | 2,3 | 2048 | 67.25 | 68.01 | 67.25 | 67.48 |
| sent | FALSE | 1,2,3 | 3072 | 66.9 | 67.01 | 66.9 | 66.9 |
| sent | FALSE | 1 | 3072 | 66.5 | 66.04 | 66.3 | 66.2 |
| sent | FALSE | 1,2 | 3072 | 66.3 | 66.04 | 66.6 | 66.4 |
| word | TRUE | 2 | 1024 | 64.79 | 64.92 | 64.79 | 64.77 |
| word | TRUE | 3 | 1024 | 64.45 | 64.99 | 64.45 | 64.57 |
| word | TRUE | 1 | 1024 | 59.19 | 59.32 | 59.19 | 59.17 |
| P-SIF (Doc2VecC) | - | - | - | **86.0** | **86.1** | **86.1** | **86.0** |
| P-SIF | - | - | - | **85.4** | **85.5** | **85.4** | **85.2** |

Table 19: Performance on Reuters classification using p-means embedding

| Power | Znorms | Dim | Prec@1 nDCG@1 | Prec@5 | nDCG@5 | Coverage Error | LRAPS | F1 -meas |
|---|---|---|---|---|---|---|---|---|
| 1,2 | FALSE | 400 | 93.07 | 36.69 | 48.96 | 7.66 | 91.73 | 76.76 |
| 1,2 | TRUE | 400 | 92.49 | 36.23 | 48.45 | 13.93 | 90.58 | 79.13 |
| 1,2,-1,+1 | FALSE | 800 | 91.9 | 35.79 | 47.91 | 9.19 | 90.04 | 75.93 |
| P-SIF (Doc2VecC) | - | - | **94.92** | **37.98** | **50.40** | **6.03** | **93.95** | **82.87** |
| P-SIF | - | - | **94.77** | **37.33** | **49.97** | **6.24** | **93.72** | **82.41** |

