# OpenReview forum: "Unsupervised Document Representation using Partition Word-Vectors Averaging"
_ICLR.cc/2019/Conference_

### Official Review · AnonReviewer2 · 2018-11-02
**review of Unsupervised Document Representation using Partition Word-Vectors Averaging**

**Rating:** 4
**Confidence:** 4

**Review:**

Paper overview: The paper extends the method proposed by Arora 2017 for sentence embeddings to longer document embeddings. The main idea is that, averaging word embedding vectors mixes all the different topics on the document, and therefore is not expressive enough. Instead they propose to estimate the topic of each word (using dictionary learning) through the $\alpha$ weights (see page 4).These weights give "how much" this word belongs to a certain topic. For every topic we compute the $\alpha$-weighted vector of the word and  concatenate them (see word topic vector formation). Finally, we apply SIF (Arora 2017) using these word embeddings on all the document.

Questions and remarks:
     1) How sensitive is the method to a change in the number of topics (k)?
    2) Please provide also the std instead of just the average performance, so that we can understand if the differences between methods are significantly meaningful

Points in favor:
   1) Good results and thorough tests
    2) Paper is easy to read and follow

Points against:
A very similar method was already proposed by Mekala 2017, as the authors acknowledge in section 7. The main difference between the two methods is that Mekala et al use GMM and the authors of the present paper use sparsity method K-SVD to define the topics.


The novelty of the paper is not enough to justify its acceptance at the conference.

---

> ### Author Response · Authors · 2018-11-07
> **Author response to AnonReviewer2**
>
> Apologies for the delay in response. We would like to thank the reviewers for evaluating our manuscript. We have tried to address all the reviewers’ concerns in a proper way and believe that our paper has improved. We would be happy to make further corrections and look forward to hearing from you soon. We respond to the questions and concerns in the following points:
>
> 1. The method is sensitive to the total number of clusters (K), which we tune using cross-validation. However, it is not very sensitive to k (#non-zero entry parameter) as the number of senses of a word is limited.
>
> 2. We will provide the std instead of only providing the average performance so that we can understand if the differences between these methods are significantly meaningful. From our preliminary experiments, we found out that std-values are in 10^-3 order. Thus the results are robust.
>
> 1. We agree that the methods draw inspiration from the SCDV (Mekala et al., 2017). However, there are significant differences between SCDV (Mekala et al., 2017) and this work:
>
> a) SCDV (Mekala et al., 2017) uses GMM whereas P-SIF uses k-svd (Arora et al. 2016) for topic modelling. It should be noted that we did not apply manual hard thresholding over final documents representation as the method in SCDV did. Because of k-svd we implicitly put the sparsity constraint during clustering. This yields several other benefits such as sparser documents thus reducing space and time complexity. For single sentence datasets (STS 12-16 and Twitter15), we found out that both GMM and k-svd work really well. GMM works slightly better as it is much easier to optimize compared to k-svd. We also noted for these datasets the total number of clusters is small. However, for datasets containing multiple sentences such as 20NewsGroup and Reuters (200-500-word documents), k-svd outperforms GMM a.k.a P-SIF outperforms SCDV. Additionally, using k-svd leads to a fewer number of total clusters and hence fewer dimensions of document vectors for better representations. As a result, the feature formation, training and prediction time are faster. We will add time and space complexity results in the paper as well.
>
> b) We used the SIF weighting and common component removal in P-SIF, whereas SCDV used tf-idf. SIF (Arora et. al. 2017) has shown the benefit of using such weighting and common component removal. We have successfully generalised SIF (Arora et. al. 2017) using the ideas from the SCDV paper. Additionally, SCDV used SGNS-initialized word vectors, whereas P-SIF used Doc2VecC-initialized word vectors. Also, in Doc2VecC the averaging is based on sentence representation and training of word vectors is done jointly with corruptions. This kind of training with corruption results in zeroing of common words' word vectors. However, since our approach is about partition based averaging such as zeroing, we can yield more robust document representation. We will add more downstream tasks which were requested by the reviewer as well. It should be noted that we have more thorough experiments on 26 STS similarity and two text classification datasets of the SCDV paper

---

> > ### Author Response · Authors · 2018-11-27
> > **Some updates on revised paper**
> >
> > We want to thank the reviewers for evaluating our manuscript. We have tried to address all the reviewers’ concerns adequately and believe that our paper has significantly improved. We have updated the paper with recent results in the Appendix.
> >
> > 1. We added a Proof. Sketch for our embedding following (Arora et. al. 2017) paper. See Appendix I.
> >
> > 2. We showed that many embeddings (word vector averaging, word mover distance, relax mover distance and P-SIF) could be expressed in the form of suitable similarity Kernels (Appendix J). We empirically showed that our embeddings outperform Word Mover Distance (ICML 2015) and Word Mover Embedding (EMNLP 2018) on eight more datasets on various classification tasks (see Appendix J Table 14). We are hopeful our results will further improve if we use P-SIF with Doc2VecC initialized word vectors instead of pre-trained GloVe vectors. We will start experiments with Doc2VecC initialized vectors and hope to finish it before the final submission.
> >
> > 3. We added some more recent baselines and compared our embedding results on Semantic Textual Similarity Tasks (Appendix K Table 15). We outperformed most benchmarks and performed similarly to state-of-the-art u-SIF embeddings. The results show that our embedding is quite competent empirically.
> >
> > 4. We added a detailed comparison of pmeans and ELMO with our embeddings (P-SIF) in Appendix L Table 16. Furthermore, we compared our embeddings (P-SIF) with p-means on Reuters. We are still running more experiments and hope to complete them before the final submission.
> >
> > Minor: Added F-Score in Table 10
> >
> > We will compress the paper to incorporate some results in the main body of paper instead of Appendix.

---

### Official Review · AnonReviewer1 · 2018-11-03
**The work is incremental even though the experimental results are good and the method is well presented.**

**Rating:** 7
**Confidence:** 4

**Review:**


Pros:
The paper shows that we could have a better document/sentence embedding by partitioning the word embedding space based on a topic model and summing the embedding within each partition. The writing and presentation of the paper are clear. The method is simple, intuitive, and the experiments show that this type of method seems to achieve state-of-the-art results on predicting semantic similarity between sentences, especially for longer sentences.

Cons:
The main concern is the novelty of this work. The method is very similar to SCDV (Mekala et al., 2017). The high-level flow figure in appendix H is nearly identical as the Figure 1 and 2 in Mekala et al., 2017. The main difference seems to be that this paper advocates K-SVD (extensively studies in Arora et al. 2016) as their topic model and SCDV (Mekala et al., 2017) uses GMM.
However, in the semantic similarity experiments (STS12-16 and Twitter15), the results actually use GMM. So I suppose the results tell us that we can achieve state-of-the-art performances if you directly combine tricks in SIF (Arora et al., 2017) and tricks in SCDV (Mekala et al., 2017).
In the document classification experiment, the improvement looks small and the baselines are not strong enough. The proposed method should be compared with other strong unsupervised baselines such as ELMo [1] and p-mean [2].

Overall:
The direction this paper explores is promising but the contributions in this paper seem to be incremental. I suggest the authors to try either of the following extensions to strengthen the future version of this work.
1. In addition to documentation classification, show that the embedding is better than the more recent proposed strong baselines like ELMo in various downstream tasks.
2. Derive some theories. One possible direction is that I guess the measuring the document similarity based on proposed embedding could be viewed as an approximation of Wasserstein similarity between the all the words in both documents. The matching step in Wasserstein is similar to the pooling step in your topic model. You might be able to say something about how good this approximation is. Some theoretical work about doing the nearest neighbor search based on vector quantization might be helpful in this direction.

Minor questions:
1. I think another common approach in sparse coding is just to apply L1 penalty to encourage sparsity. Does this K-SVD optimization better than this L1 penalty approach?
2. How does the value k in K-SVD affect the performances?
3. In Aorora et al. 2016b, they constrain alpha to be non-negative. Did you do the same thing here?
4. How important this topic modeling is? If you just randomly group words and sum the embedding in the group, is that helpful?
5. In Figure 2, I would also like to see another curve of performance gain on the sentences with different lengths using K-SVD rather than GMM.

Minor writing suggestions:
1. In the 4th paragraph of section 3, "shown in equation equation 2", and bit-wise should be element-wise
2. In the 4th paragraph of section 4, I think the citation after alternating minimization should be Arora et al. 2016b and Aharon et al. 2006 rather than Arora et al., 2016a
3. In the 2nd paragraph of section 6.1, (Jeffrey Pennington, 2014) should be (Pennington et al., 2014). In addition, the author order in the corresponding Glove citation in the reference section is incorrect. The correct order should be Jeffrey Pennington, Richard Socher, Christopher D. Manning.
4. In the 3rd paragraph of section 6.1, "Furthermore, Sentence"
5. In the 6th paragraph of section 6.1, I thought skip-thoughts and Sent2Vec are unsupervised methods.
6. In Table 2 and 3, it would be easier to read if the table is transposed and use the longer name for each method (e.g., use skip-thought rather than ST)
7. In Table 2,3,4,5, it would be better to show the dimensions of embedding for each method
8. Table 10 should also provide F1
9. Which version of GMM is used in STS experiment? The one using full or diagonal covariance matrix?


[1] Peters, M. E., Neumann, M., Iyyer, M., Gardner, M., Clark, C., Lee, K., & Zettlemoyer, L. (2018). Deep contextualized word representations. NAACL
[2] Rücklé, A., Eger, S., Peyrard, M., & Gurevych, I. (2018). Concatenated p-mean Word Embeddings as Universal Cross-Lingual Sentence Representations. arXiv preprint arXiv:1803.01400.

---

> ### Author Response · Authors · 2018-11-07
> **Author response to AnonReviewer1**
>
> Apologies for the delay in response. We would like to thank the reviewer for evaluating our manuscript. We have tried to address the reviewer concerns in a proper way and believe that our paper has improved. We would be happy to make further corrections and look forward to hearing from you soon. We respond to the questions and concerns in the following points:
>
> 1. We agree that the methods draw inspiration from the SCDV (Mekala et al., 2017). However, there are major differences between SCDV (Mekala et al., 2017) and this work:
>
> a) SCDV (Mekala et al., 2017) uses GMM whereas P-SIF uses k-svd (Arora et al. 2016) for topic modelling. It should be noted that we did not apply manual hard thresholding over final documents representation as the method in SCDV did. Because of k-svd we implicitly put the sparsity constraint during clustering. This yields several other benefits such as sparser documents thus reducing space and time complexity. For single sentence datasets (STS 12-16 and Twitter15), we found out that both GMM and k-svd work really well. GMM works slightly better as it is much easier to optimize compared to k-svd. We also noted for these datasets the total number of clusters is small. However, for datasets containing multiple sentences such as 20NewsGroup and Reuters (200-500-word documents), k-svd outperforms GMM a.k.a P-SIF outperforms SCDV. Additionally, using k-svd leads to a fewer number of total clusters and hence fewer dimensions of document vectors for better representations. As a result, the feature formation, training and prediction time are faster. We will add time and space complexity results in the paper as well.
>
> b) We used the SIF weighting and common component removal in P-SIF, whereas SCDV used tf-idf. SIF (Arora et. al. 2017) has shown the benefit of using such weighting and common component removal. We have successfully generalized SIF (Arora et. al. 2017) using the ideas from the SCDV paper. Additionally, SCDV used SGNS-initialised word vectors, whereas P-SIF used Doc2VecC-initialised word vectors. Also, in Doc2VecC the averaging is based on sentence representation and training of word vectors is done jointly with corruptions. This kind of training with corruption results in zeroing of common words' word vectors. However, since our approach is about partition based averaging such as zeroing, we can yield more robust document representation. We will add more downstream tasks which were requested by the reviewer as well. It should be noted that we have more thorough experiments on 26 STS similarity and 2 text classification datasets of the SCDV paper (28 in total).
>
> The proposed baselines in document classification are taken from a very recently published paper (2017-2018). As suggested by reviewer we will add more baselines such as the Elmo [1] and p-mean [2] for text classification. We didn't find the baselines for our reported datasets, but we have found the code to run on our datasets.
>
> Minor Questions:
> 1. Yes, but directly optimising the L1 is an NP-hard objective. So k-svd does an alt-min (Arora et al. 2016b and Aharon et al. 2006) between clustering and thresholding to achieve the require sparsity.
> 2. We keep the k small, unlike other normal k-svd applications as we know that in text each word has a very limited number (< 5) of total senses. We use the procedure similar to (Arora et al. 2016b) to choose the optimal k. For our experiments, we take k equal to the total_clusters of clusters (K) divided by 10, which is a good approximation.
> 3. Yes, if we randomly average words in same clusters and analyze the top dominant words in the clusters, we observe words with similar meanings are close to each other. A similar observation was reported by (Mekala et al., 2017) and (Arora et al. 2016b).
> 4. We will plot the point. Our institution is that initially for small length documents GMM and K-svd perform equally well (GMM may be slightly better due to easier optimization) but later for long length documents k-svd will easily outperform GMM with a fewer lesser number of clusters as reported in the text classification.
> 5. We used the full covariance matrix of the GMM in our paper.
>
> We thank the reviewer for providing helpful directions for theoretical derivations
> We were having doubts about how to proceed with the theoretical analyzes. The idea of measuring the document similarity based on P-SIF embedding and viewing it as an approximation of Wasserstein similarity between all the words in both documents seem interesting. As stated, the matching step in Wasserstein is indeed similar to the pooling step in our topic model. We will also delve into the theoretical work on doing the nearest neighbour search on vector quantizations. Thanks for your suggestion. We pinned down atleast one relevant paper: http://proceedings.mlr.press/v37/kusnerb15.pdf .
>
> Minor writing corrections
> We made the correction suggested in the revised version. We didn't transpose the table due to page limitation.

---

> > ### Comment · AnonReviewer1 · 2018-11-07
> > **Sounds good**
> >
> > Thanks for the responses.
> >
> > I think one of my concerns is not addressed very well. When you concatenate the average word embedding from different clusters, your number of dimension increases. If you have many clusters, your number of dimension might be much larger than other types of embedding. It might not be very fair to compare the performance without showing the number of dimensions in each table.
> > My 4th question in the minor questions section is also about this concerns, but I do not understand your response. My question is that if you do not perform clustering, you just randomly assign/hash each word to a group/bin and then average the words inside the group and bin. If this simple baseline could also improve the performance on those STS or multiclass classification problem, it might imply that P-SIF improves from SIF mostly because it uses more dimensions.
> >
> > By the way, I think you also forgot to reply my 3rd minor question (the non-negativity constraint).
> >
> > Overall:
> > If you could clarify the above dimension concern and either
> > 1) have some significant theoretical contributions or
> > 2) show that P-SIF as pretraining is better than ELMo and p-mean on document classification in Table 4 and 5, and on 1 more downstream task (do not include semantic similarity prediction),
> > I will vote for acceptance.

---

> > > ### Author Response · Authors · 2018-11-15
> > > **Author responce to AnonReviewer1**
> > >
> > > Apologies for the delay in response. Thanks for your patience.
> > >
> > > It is true that the average size of our embedding is much larger than many other embeddings in the baseline but P-SIF embedding and other baselines embeddings are all derived from similar lower dimensional (200D) unsupervised trained word vectors. The difference in performance is due to the difference in composition operation rather than higher dimensionality. The reported baseline is the best performance embedding taken from the original paper which is reported after fine-tuning. We did not achieve any improvements in the baselines with further increase in the embedding dimensions, e.g. the tf-idf weighted average word vectors with 2000 dimensions only yields 0.2% improvement on 20NewsGroup dataset.
> > >
> > > If we randomly assign/hash each word to a random group/bin (with equal bin probability) and then average the words inside each group/bin, we wouldn't get any better results than the averaging of the Doc2VecC initialized word vectors directly without any hashing (reported as the Doc2veC baseline in the paper).
> > >
> > > We present results for three random runs on the 20NewsGroup dataset. The dimensions of the embedding are similar to our embedding, i.e., 200 x 40 = 8000. You can access the results here: https://goo.gl/KbGR81 (Table 14)
> > >
> > > As expected, the result is similar to that of the Doc2VecC average reported in our paper.
> > >
> > > Our main idea of clustering the word-vectors was to ensure that similar meaning words get mapped/hashed/assigned to a similar cluster. Averaging of any linear combination of word-vectors within this cluster will always keep the resultant vector close to the original cluster. If we randomly assign/hash each word to a group/bin and then average the words inside each bin, this means that we will average very dissimilar words' word-vectors with different meanings. The produced average vectors will not necessarily fall in the same cluster. Thus, we end up with the averaging of random words' word vectors which results in random word vectors in which the meanings are dissimilar to the original words.
> > >
> > > We did not set any explicit constraint for the positivity of the alpha_i. Arora et. al. 2016b also did not put any non-negativity constraints. In fact, we observe sparse coding procedures assigned non-negative values to most coefficients alpha_i, even if they are unrestricted. Perhaps, this is because of the fact that the appearances of a word are best explained with the help of the theme/topic being used to generate it, rather than the theme/topic is not being used. A similar observation was reported by Arora et. al. 2016b, please refer to sec 5 Experiments with Atom of Discourse 5th paragraph. We have mentioned this point in our paper, please refer sec 5 PSIF Discussion, Sparse Dictionary Learning vs. Fuzzy Clustering.
> > >
> > > As suggested by the reviewer, we ran experiments on 20NewsGroup dataset for the ELMO and P-mean Univeral Embedding baselines. For p-mean we tried various values and combinations of p (as reported), we also checked the effect of z-normalization and tf-idf weighting.  You can access the results here: https://goo.gl/KbGR81 (Table 15)
> > >
> > > For EMLO, we ran the experiments on 20NewsGroup using the allennlp package (https://allennlp.org/elmo) with the original model. We used the default options and weight file (code package: https://github.com/allenai/allennlp/blob/master/tutorials/how_to/elmo.md)
> > >
> > > We tried averaging and concatenation of multiple layers. We also run the experiments by only using the top layer. In addition, we tried our experiments based on the whole document words as context as well as each sentence as a context.  You can access the results here: https://goo.gl/KbGR81 (Table 16)
> > >
> > > We are still running ELMO and P-means experiments on the Reuters dataset. It is taking more time than what was expected because we are employing a one vs rest multi-label classification approach (~90 classifiers) and the documents also have more words than 20NewsGroup.
> > >
> > > For the STS 12-16 task, we will include and discuss some more baseline in Table 2 from the following two recent survey papers: https://goo.gl/aLTiS5 and https://goo.gl/PjjBRD . P-SIF is able to outperform most of these baselines. We were able to outperform most of the baseline and perform similarly to State-of-the-art u-SIF from http://aclweb.org/anthology/W18-3012 .  You can access the results here:
> > >  https://goo.gl/KbGR81 (Table 17)
> > >
> > > We will update the submission draft with new results in the next revision.
> > >
> > > We are working on a strong and significant proof while taking inspiration from the generative gaussian random walk model in (Arora et. al. 2016b) and simple random walk model in (Arora et. al. 2016a) for word vectors representations and its application to prove SIF averaging model as in (Arora et. al. 2017).  We are still improving the proof. You can access our proof sketch here: https://goo.gl/fHn1rN .

---

> > > > ### Comment · AnonReviewer1 · 2018-11-15
> > > > **The experimental results look promising but theoretical analysis is not significant yet**
> > > >
> > > > If I understand it correctly, the proof sketch basically just says that we can assign each word to cluster with some probability (Arora et. al. 2016b). After fixing the probability, we could view the document as K bags of words and use the same analysis in Arora et. al. 2017 to justify the weighted average behavior within each bag. This is not significant. One example of significant results is to show that this two-stage process (estimating co-efficient and estimating embedding) actually optimizes a unified objective function (in an approximated way?).
> > > >
> > > > Minor things in your proof sketch:
> > > > I think your notations are not consistent and confusing. For example, you sometimes use m to represent K. alpha(w) in (6,7) and alpha seems to be the same thing, but alpha_(w,j) is different. Your p(w) seems to mean global frequency of w in the whole document sometimes and the frequency of w in each bag/partition sometimes. Do arrow{c_j} in (9,10,12) and arrow{V_{c_j}} mean the same thing? Your footnote 11 comes from Arora et. al. 2017, but I think it makes more sense to use argmax instead of max here.
> > > >
> > > >
> > > > P-SIF is much better than ELMo and p-mean in 20NewsGroup, which is pretty nice. I look forward to the results on Reuters and 1 more downstream task.

---

> > > > > ### Author Response · Authors · 2018-11-27
> > > > > **Author responce to AnonReviewer1**
> > > > >
> > > > > Apologies for the delay in response. We want to thank the reviewer for evaluating our manuscript. We have tried to address all the reviewers’ concerns adequately and believe that our paper has significantly improved. We have updated the paper with recent results in the Appendix.
> > > > >
> > > > > Corrected minor points in Proof Sketch (Appendix I)
> > > > >
> > > > > We showed that many embeddings (word vector averaging, word mover distance, relax mover distance and P-SIF) could be expressed in the form of suitable similarity Kernels (Appendix J). We empirically showed that our embeddings outperform Word Mover Distance (ICML 2015) and Word Mover Embedding (EMNLP 2018) on eight more datasets on various classification tasks (see Appendix J Table 14). We are hopeful our results will further improve if we use P-SIF with Doc2VecC initialized word vectors instead of pre-trained GloVe vectors. We will start experiments with Doc2VecC initialized vectors and hope to finish it before the final submission.
> > > > >
> > > > > We added some more recent baselines and compared our embedding results on Semantic Textual Similarity Tasks (Appendix K Table 15). We outperformed most benchmarks and performed similarly to state-of-the-art u-SIF embeddings. The results show that our embedding is quite competent empirically.
> > > > >
> > > > > We added a detailed comparison of pmeans and ELMO with our embeddings (P-SIF) in Appendix L Table 16. Furthermore, we compared our embeddings (P-SIF) with p-means on Reuters. We are still running more experiments and hope to complete them before the final submission.
> > > > >
> > > > > We were unable to provide significant theoretical results despite trying very hard. We will compress the paper to incorporate some results in the main body of paper instead of Appendix.

---

> > > > > > ### Comment · AnonReviewer1 · 2018-11-27
> > > > > > **The new results are strong**
> > > > > >
> > > > > > I believe the revised version has added sufficient experiments to become a good empirical paper. For example, authors show that P-SIF very significantly outperforms ELMO and p-mean in 20NewsGroup and also outperforms WME and many other baselines in several downstream tasks in Table 14.
> > > > > > The new good results make me change my vote to accept as I promise originally.
> > > > > > I am glad to see the authors include a very recent approach, u-SIF, into the table. I checked the approach from u-SIF and I think their method is orthogonal to this paper, so the authors should be able to combine u-SIF and p-SIF to achieve the truly state-of-the-art results on sentence similarity tasks (I strongly recommend authors to actually try that).
> > > > > >
> > > > > > I think the math in Appendix J is either very confusing or incorrect. Before the final publication, authors need to define the notations more clearly and show more derivation steps which lead to the conclusions (K1 is the kernel when each document is represented by average word embeddings, K2 is the kernel of your method, and K5 is the kernel of word mover distance).

---

> > > > > > > ### Author Response · Authors · 2018-11-30
> > > > > > > **Author responce to AnonReviewer1**
> > > > > > >
> > > > > > > Thank you for changing the vote to acceptance
> > > > > > >
> > > > > > > Yes, u-SIF is orthogonal to our paper. As per the reviewer's suggestion, we will combine u-SIF and p-SIF to further improve the results on the sentence similarity tasks.
> > > > > > >
> > > > > > > As per reviewer's suggestion we updated the notation and added some derivation steps which connect the kernel and embedding in Appendix J. You can access the updates here https://goo.gl/GD2Hia

---

### Official Review · AnonReviewer3 · 2018-11-06
**A very well written paper with solid technical contribution.**

**Rating:** 6
**Confidence:** 3

**Review:**

A very well written paper with solid technical contribution. The impact to the community might be incremental.

Pros:
1. I enjoyed reading the paper, very well written, clean, and organized.
2. Comprehensive literature survey, the authors provided both enough context for the readers to digest the paper, and well explained how this work is different from the existing literature.
3. Conducted extensive experiments.

Cons (quibbles):
Experiments:
The authors didn't compare the proposed method against topic model (vanilla LDA or it’s derivatives discussed in related work). Because most topic models could generate vector representation for document too, and it's interesting to learn additional benefit of local context provided by the word2vec-like model.

Methodology:
1. About hyperparameters:
a. Are there principled way/guideline of finding sparsity parameters k in practice?
b. How about the upper bound m (or K, the authors used both notation in the paper)?

2. About scalability:
How to handle such large sparse word vectors, as it basically requires K times more resource compared to vanilla word2vec and it's many variants, when it’s used for other large scale downstream cases? (see all the industrial use cases of word vector representations)

3. A potential alternative model: The motivation of this paper is that each word may belong to multiple topics, and one can naturally extend the idea to that "each sentence may belong to multiple topics". It might be useful to apply dictionary learning on sentence vectors (e.g., paraphrase2vec) instead of on word vectors, and evaluate the performance between these two models. (future work?)

Typos:
The authors mentioned that "(Le & Mikolov, 2014) use unweighted averaging for representing short phrases". I guess the authors cited the wrong paper, as in that paper Le & Mikolov proposed PV-DM and PV-DBOW model which treats each sentence as a shared global latent vector (or pseudo word).

---

> ### Author Response · Authors · 2018-11-07
> **Author response to AnonReviewer3**
>
> Apologies for the delay in response. We would like to thank the reviewers for evaluating our manuscript. We have tried to address all the reviewers’ concerns in a proper way and believe that our paper has improved. We would be happy to make further corrections and look forward to hearing from you soon. We respond to the questions and concerns in the following points:
>
> Experiments: We have compared our approach with the LDA topic modelling representation for the 20NewsGroup dataset (see Table 5). We will add the LDA baseline for the Reuters dataset as well.
>
> Methodology: The method is sensitive to the total number of clusters (K), which we tune using cross-validation. However, it is not very sensitive to k (#non-zero entry parameter) as the number of senses of words is limited. For our experiments, we take k equal to the total_clusters of clusters (K) divided by 10, which is a good approximation. The upper bound depends on two things a) average length of documents in the corpus b) number of diverse different themes (or topics) in the corpus. For both the long document corpus and for the diverse themes corpus, we require larger K. For all our experiments we tune the value of the hyper-parameter K.
>
> Scalability: The dimensions of the word topic vectors are K x d (K= #topics, d= #word2vec dimensions), which is larger than the vanilla word vector dimension (d). However, for each word only k (sparsity parameter) values are active, therefore our representation is very sparse. Thus, we can store them in a sparse format and the can perform a sparse operation for speed up. For downstream tasks where a continuous low dimensional representation is required, we can perform a lower dimension manifold learning over the learnt word topic vectors. We are currently delving into the details of the effective approaches to learn this lower dimensional manifold for higher dimension sparse vectors.
>
> A potential alternative model: We thank the reviewer for proposing an alternative model. We will investigate more about the proposed idea. Some immediate drawbacks in the proposed model are a) the assumption that each sentence consists of only single topics and b) each sentence is represented in the same d dimension vector space as the vanilla word vectors. In P-SIF we did not make these assumptions. As suggested, we will compare our results with the proposed baseline
>
> We have corrected the typos.

---

> > ### Author Response · Authors · 2018-11-27
> > **Some updates on revised paper**
> >
> > We want to thank the reviewers for evaluating our manuscript. We have tried to address all the reviewers’ concerns adequately and believe that our paper has significantly improved. We have updated the paper with recent results in Appendix.
> >
> > 1. We added a Proof. Sketch for our embedding following (Arora et. al. 2017) paper. See Appendix I.
> >
> > 2. We showed that many embeddings (word vector averaging, word mover distance, relax mover distance and P-SIF) could be expressed in the form of suitable similarity Kernels (Appendix J). We empirically showed that our embeddings outperform Word Mover Distance (ICML 2015) and Word Mover Embedding (EMNLP 2018) on eight more datasets on various classification tasks (see Appendix J Table 14). We are hopeful our results will further improve if we use P-SIF with Doc2VecC initialized word vectors instead of pre-trained GloVe vectors. We will start experiments with Doc2VecC initialized vectors and hope to finish it before the final submission.
> >
> > 3. We added some more recent baselines and compared our embedding results on Semantic Textual Similarity Tasks (Appendix K Table 15). We outperformed most benchmarks and performed similarly to state-of-the-art u-SIF embeddings. The results show that our embedding is quite competent empirically.
> >
> > 4. We added a detailed comparison of pmeans and ELMO with our embeddings (P-SIF) in Appendix L Table 16. Furthermore, we compared our embeddings (P-SIF) with p-means on Reuters. We are still running more experiments and hope to complete them before the final submission.
> >
> > Minor: Added F-Score in Table 10
> >
> > We will compress the paper to incorporate some results in the main body of paper instead of Appendix.

---

### Author Response · Authors · 2018-11-27
**Author updates for Area Chair**

We have tried to address all the reviewers’ concerns adequately and believe that our paper has significantly improved. We have updated the paper with recent results in the Appendix.

1. We added a Proof. Sketch for our embedding following (Arora et. al. 2017) paper. See Appendix I.

2. We showed that many embeddings (word vector averaging, word mover distance, relax mover distance and P-SIF) could be expressed in the form of suitable similarity Kernels (Appendix J). We empirically showed that our embeddings outperform Word Mover Distance (ICML 2015) and Word Mover Embedding (EMNLP 2018) on eight more datasets on various classification tasks (see Appendix J Table 14). We are hopeful our results will further improve if we use P-SIF with Doc2VecC initialized word vectors instead of pre-trained GloVe vectors. We will start experiments with Doc2VecC initialized vectors and hope to finish it before the final submission.

3. We added some more recent baselines and compared our embedding results on Semantic Textual Similarity Tasks (Appendix K Table 15). We outperformed most benchmarks and performed similarly to state-of-the-art u-SIF embeddings. The results show that our embedding is quite competent empirically.

4. We added a detailed comparison of pmeans and ELMO with our embeddings (P-SIF) in Appendix L Table 16. Furthermore, we compared our embeddings (P-SIF) with p-means on Reuters. We are still running more experiments and hope to complete them before the final submission.

Minor: Added F-Score in Table 10

We think our paper is theoretically and empirically strong and hence sufficient for publication at ICLR 2019. We specially thanks reviewer 2 for useful suggestions (& patience) which help us in significantly improving our paper.

We will compress the paper to incorporate some results in the main body of paper instead of the Appendix.

---

### Meta-Review · Area_Chair1 · 2018-12-14
**borderline paper**

**Confidence:** 4
**Recommendation:** Reject

**Metareview:**

This paper proposes a document classification algorithm based on partitioned word vector averaging.
I agree with even the most positive reviewer. More experiments would be good. This is a very developed old area.

---

> ### Author Response · Authors · 2018-12-21
> **Author responce to Area Chair**
>
> We want to draw area chair's attention towards the new set of experiments and baselines, to incorporate reviewers opinions (especially reviewer 1). We have now compared our classification on 20NewsGroup with ELMO and p-means (Table 4) and Reuters on pmeans (Table 5). We have also compared our approach on STS tasks with several recent strong baselines (Table 2, 3 & 15 (Appendix K) and also experimented on other text classification datasets with several benchmarks (Table 14), to show P-SIF's effectiveness. With all due respect to the area chair's opinion, we think the paper contains enough empirical support for the P-SIF embedding. We request the area chair to reconsider the decision with all these points in mind.